# Functional and structural insights into activation of TRPV2 by weak acids

Ferdinand M Haug[1,7], Ruth A Pumroy[2,7], Akshay Sridhar[3], Sebastian Pantke [1], Florian Dimek[1], Tabea C Fricke [1], Axel Hage[1], Christine Herzog[1], Frank G Echtermeyer[1], Jeanne de la Roche [4], Adrian Koh[5], Abhay Kotecha[5], Rebecca J Howard [6], Erik Lindahl[3,6], Vera Moiseenkova-Bell [2✉] & Andreas Leffler [1✉]

## Abstract

Transient receptor potential (TRP) ion channels are involved in the surveillance or regulation of the acid-base balance. Here, we demonstrate that weak carbonic acids, including acetic acid, lactic acid, and $CO_2$ activate and sensitize TRPV2 through a mechanism requiring permeation through the cell membrane. TRPV2 channels in cell-free inside-out patches maintain weak acid-sensitivity, but protons applied on either side of the membrane do not induce channel activation or sensitization. The involvement of proton modulation sites for weak acid-sensitivity was supported by the identification of titratable extracellular (Glu495, Glu561) and intracellular (His521) residues on a cryo-EM structure of rat TRPV2 (rTRPV2) treated with acetic acid. Molecular dynamics simulations as well as patch clamp experiments on mutant rTRPV2 constructs confirmed that these residues are critical for weak acid-sensitivity. We also demonstrate that the pore residue Glu609 dictates an inhibition of weak acid-induced currents by extracellular calcium. Finally, TRPV2-expression in HEK293 cells is associated with an increased weak acid-induced cytotoxicity. Together, our data provide new insights into weak acids as endogenous modulators of TRPV2.

**Keywords** Acidosis; Modulation Site; Cryo-EM; Protons; Transient Receptor Channel V2
**Subject Categories** Membranes & Trafficking; Molecular Biology of Disease; Structural Biology

## Introduction

The transient receptor potential vanilloid (TRPV) subfamily belongs to the large superfamily of transient receptor potential channels. TRPV1-TRPV6 are polymodal cation channels that are activated by numerous endogenous and exogenous ligands as well as by physical stimuli,
including heat, cold, and pressure (Rosenbaum et al, 2022). While TRPV2 has long been the least well understood member of the TRPV subfamily, there is growing evidence for an eminent relevance of TRPV2 in several physiological processes. TRPV2 has been implicated in neuronal outgrowth (Cohen et al, 2015; Shibasaki et al, 2010; Sugio et al, 2017), mechano-sensitivity of sensory neurons (Katanosaka et al, 2018), myocardial structure and function (Entin-Meer and Keren, 2020; Katanosaka et al, 2014), innate immunity (Link et al, 2010), brown fat thermogenesis (Sun et al, 2016), endocrine secretion (Hisanaga et al, 2009) and endometrial development (De Clercq et al, 2017). From a pathophysiological perspective, TRPV2 not only seems to be highly relevant for growth and invasiveness of different tumors (Conde et al, 2021; Huang et al, 2022; Shoji et al, 2023; Siveen et al, 2020), but also for the permeation of several virus subtypes through the cell membrane of myeloid cells (Guo et al, 2022). A significant challenge in studying TRPV2 is the lack of potent and specific agonists and antagonists. While 2-aminoethoxydiphenyl borate (2-APB), cannabidiol (CBD) and probenecid are commonly used TRPV2 activators, they have many off target effects, including modulation of other TRPV channels (Chung et al, 2004; Etemad et al, 2022; Zhang et al, 2022). High heat exceeding >50 °C activates rodent TRPV2 (Caterina et al, 1999), but human TRPV2 is insensitive to heat (Neeper et al, 2007). This species difference raises the question of whether heat-induced activation of TRPV2 is endogenously relevant, and mice lacking TRPV2 do not have a phenotype in regard to heat-induced nociceptive behavior (Park et al, 2011). TRPV2 was also reported to be gated by mechanical or osmotic stress (Mihara et al, 2010; Muraki et al, 2003). More recent studies have identified putative endogenous modulators of TRPV2, including reactive oxygen species (ROS) (Fricke et al, 2019), a tyrosine-dependent phosphorylation (Mo et al, 2022) and cholesterol (Su et al, 2023).

Acidosis is a common endogenous modulator of several TRP channels, i.e., protons act as agonists or inhibitors on different TRP-channel subtypes. TRPV1 is directly activated and sensitized by extracellular protons (Caterina et al, 1997; Tominaga et al, 1998). The extracellular residues Glu600 and Glu648 have been validated as critical extracellular proton modulation sites of TRPV1 (Jordt et al,

[1]Department of Anesthesiology and Intensive Care Medicine, Hannover Medical School, 30625 Hannover, Germany. [2]Department of Systems Pharmacology and Translational Therapeutics, Perelman School of Medicine, University of Pennsylvania, Philadelphia, USA. [3]Department of Applied Physics, Science for Life Laboratory, KTH Royal Institute of Technology, Stockholm, Sweden. [4]Institute for Neurophysiology, Hannover Medical School, Hannover, Germany. [5]Thermo Fisher Scientific, Eindhoven, The Netherlands. [6]Department of Biochemistry and Biophysics, Science for Life Laboratory, Stockholm University, Stockholm, Sweden. [7]These authors contributed equally: Ferdinand M Haug, Ruth A Pumroy. ✉E-mail: vmb@pennmedicine.upenn.edu; leffler.andreas@mh-hannover.de

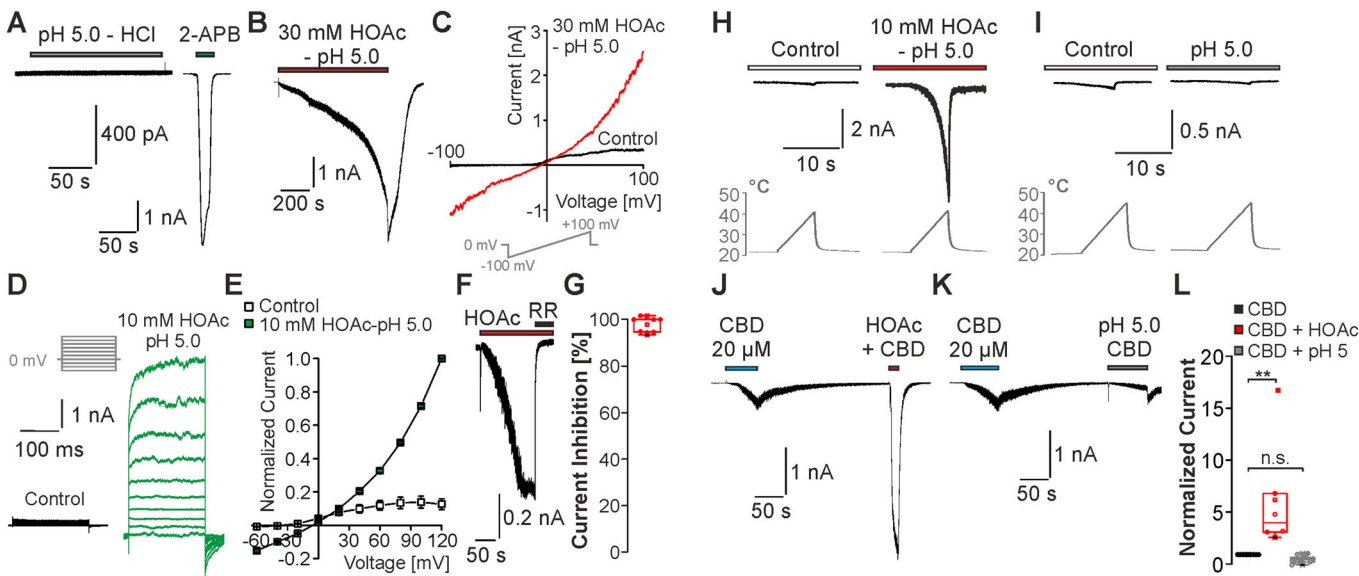

**Figure 1. rTRPV2 is activated by acetic acid.**

(A) Representative current trace from a rTRPV2-expressing cell displaying that pH 5.0 titrated with HCl fails to induce inward currents. Note that this cell produced a large 2-APB-induced current, e.g., it expressed rTRPV2. (B) 30 mM HOAc titrated to pH 5.0 induces slowly activating and fully reversible inward currents in cells expressing rTRPV2. Cells were held at −60 mV. (C) Typical membrane currents in control solution or 30 mM HOAc at pH 5.0 monitored during a 500 ms long voltage ramp ranging from −100 mV to 100 mV. (D) Voltage-dependent membrane currents evoked by 200 ms long pulses from −60 to 120 mV applied in steps of 20 mV. Currents were evoked in control solution and in 10 mM HOAc at p 5.0. (E) Current–voltage plots of experiments performed in (C). Peak current amplitudes were normalized to the amplitude evoked at 120 mV. Data are mean ± SEM, $n = 6$. (F) Typical inward currents evoked by 30 mM HOAc and pH 5.0 inhibited by 10 µM ruthenium red (RR). (G) Box diagrams with dot plots displaying the degree of current reduction by ruthenium red. (H) Traces displaying HOAc-induced sensitization of rTRPV2 to heat. (I) Traces displaying the lack of effect of pH 5.0 titrated with HCl on heat-sensitivity of rTRPV2. (H, I) Solutions were applied as "heat ramps" from room temperature to 40–45 °C. Note that heat-evoked currents were only observed in the presence of HOAc. (J, K) Current traces on rTRPV2-expressing cells with two consecutive applications of 20 µM CBD. For the 2nd application, CBD was combined with 10 mM HOAc at pH 5.0 (J) or pH 5.0 with HCl (K). (L) Box diagrams with dot plots displaying normalized current amplitudes of currents in induced by CBD (set as 1), CBD + HOAc ($n = 6$) or CBD + pH 5.0 ($n = 9$). **$P < 0.01$. One-way ANOVA, Bonferroni post hoc test. (G, L) The box denotes the 50th percentile (median) as well as the 25th and 75th percentile. The whiskers mark the 5th and 95 percentiles. Data points beyond the whiskers are outliers. Source data are available online for this figure.

2000; Zhang et al, 2021), but further residues were suggested to regulate activation of TRPV1 by protons as well (Aneiros et al, 2011; Ryu et al, 2007; Wang et al, 2010). Less attention has been drawn to the reports demonstrating that TRPV1 is inhibited by weak acids, and that protons induce a direct channel inhibition when applied to the cytosolic side (Chung et al, 2011; de la Roche et al, 2016; Wang et al, 2011). In contrast, TRPV3, as well as the irritant receptor TRPA1, are activated by weak acids (Cao et al, 2012; Gao et al, 2016; Wang et al, 2010, 2011). While both ion channels seem to be directly activated by intracellular protons, no distinct proton modulation sites have yet been identified for TRPV3 or TRPA1.

Given that TRPV2 displays a high sequence homology as well as overlapping functional properties with TRPV1 and TRPV3, it is notable that previous studies have shown that both extracellular and intracellular protons fail to activate TRPV2 (Cao et al, 2012; Caterina et al, 1999; Gao et al, 2016). In this study, we report on an as yet unappreciated property of TRPV2 allowing weak acids to activate and sensitize the channel. By combining patch clamp electrophysiology, cryo-EM in lipid nanodiscs, and molecular dynamics (MD) simulations, we demonstrate that previously identified intracellular binding sites for 2-APB are also critical for weak acid-sensitivity of rat TRPV2 (Pumroy et al, 2022). In brief, our report describes a detailed characterization of a novel and likely physiologically relevant acid-sensitivity of TRPV2.

# Results

We first challenged HEK293T cells expressing rat TRPV2 (rTRPV2) with pH 5.0 titrated either with the strong hydrochloric acid (HCl), or the weak acetic acid (HOAc). While HCl fully dissociates and is therefore membrane-impermeable, the undissociated form of HOAc can permeate through the membrane and induce intracellular acidosis (Fig. EV1A). Solution titrated to pH 5.0 with HCl did not induce any inward currents except for the rapidly activating and inactivating currents known to be mediated by endogenously expressed ASIC1a channels in some cells (Figs. 1A and EV1B) (Gunthorpe et al, 2001). In contrast, application of 30 mM HOAc titrated to pH 5.0 evoked slowly activating inward currents that were reversible upon washout (Fig. 1B). These currents were not observed in non-transfected cells (Fig. EV1B). In voltage ramps ranging from −100 to +100 mV, 30 mM HOAc at pH 5.0 induced large membrane currents with a prominent inward and outward rectification (Fig. 1C, $n = 5$). This strong outward rectification was also observed on smaller membrane currents evoked by 10 mM HOAc at pH 5.0 examined by means of 200 ms long voltage steps applied from −60 to +120 mV in steps of 20 mV (Fig. 1D,E, $n = 6$). HEK293 cells were recently described to endogenously express proton-activated chloride channels (PAC or TMEM206) which produce large proton-activated outward chloride

currents at positive membrane potentials (Ullrich et al, 2019; Yang et al, 2019). In agreement with this, pH 5.0 titrated with HCl evoked large PAC-like outward currents that were blocked by the PAC-inhibitor pregnenolone sulfate in non-transfected HEK293T cells (Fig. EV1C,D). Surprisingly, these PAC-like currents were almost completely inhibited by 10 or 30 mM HOAc at pH 5.0 (Fig. EV1E,F), and no outwardly rectifying currents were evoked by 10 mM HOAc at pH 5.0 when non-transfected cells were examined with voltage steps (Fig. EV1G,H). Furthermore, the TRP-channel inhibitor ruthenium red (10 μM) strongly inhibited the HOAc-evoked inward currents in cells expressing rTRPV2 (Fig. 1F,G, $n = 9$, $98 \pm 1\%$ inhibition). These data allow us to conclude that only TRPV2 accounts for the observed HOAc-induced inward currents. This notion also seems to apply for HOAc-induced outward currents recorded at positive potentials, but it is possible that endogenous PAC channels to a limited degree contribute to these currents.

We next asked if HOAc also induces potentiation of TRPV2-mediated currents induced by other stimuli. The most potent TRPV2-agonist available to date is 2-APB. However, the potency of 2-APB on TRPV1, TRPV2, and TRPV3 was demonstrated to be increased by protons due to a modification of its chemical structure (Gao et al, 2016). Accordingly, we observed that pH 5.0 titrated with either HCl or 10 mM HOAc induced a strong potentiation of 2-APB-induced currents (Fig. EV1I–K). These data either implicate that 2-APB may not be suitable to study effects induced by weak acids, or that extracellular protons sensitize TRPV2. Therefore, we explored the effects of pH 5.0 titrated with HCl as well as 10 mM HOAc at pH 5.0 when co-applied with heat or CBD. Because rTRPV2 exhibits a strong auto-sensitization when repeatedly activated by heat (Caterina et al, 1999; Liu and Qin, 2016), we applied subthreshold heat ramps (~45 °C) that do not induce activation of rTRPV2 in control solution (Fig. 1H) (Fricke et al, 2019; Mo et al, 2022; Zhang et al, 2022). Application of 10 mM HOAc at pH 5.0 induced small inward currents that exhibited a strong potentiation upon application of heat ramps (Fig. 1H, $n = 6$). In contrast, pH 5.0 titrated with HCl did not provoke heat-induced currents (Fig. 1I, $n = 5$). HOAc also induced a strong potentiation of inward currents induced by 20 μM CBD in rTRPV2-expressing cells (Fig. 1J,L, $n = 6$, one-way ANOVA with Bonferroni correction, DF = 2, $F$ value: 9.86266). In contrast, pH 5.0 titrated with HCl did not potentiate CBD-induced currents (Fig. 1K,L, $n = 9$, $P = 1$). Together, these data demonstrate that HOAc can both activate and sensitize TRPV2.

In order to determine if gating of rTRPV2 by HOAc is an effect limited to HEK293T cells, we examined neuroblastoma ND7/23 cells transfected with rTRPV2. HOAc induced activation of inward currents as well as sensitization of heat-evoked currents on rTRPV2 expressed in ND7/23 cells (Appendix Fig. 1A,B). As TRPV2 displays considerable species-specificities in regard to activation by heat and 2-APB (Neeper et al, 2007), we also examined the mouse (m) and human (h) orthologues of TRPV2. Similar to rTRPV2, cells expressing mTRPV2 produced prominent inward as well as outward currents induced by 30 mM HOAc at pH 5.0 (Appendix Fig. 1C,D). They also exhibited large heat-evoked currents when challenged with 10 mM HOAc at pH 5.0 (Appendix Fig. 1E). In contrast, cells expressing hTRPV2 failed to produce inward currents when treated with 30 mM HOAc at pH 5.0 (Appendix Fig. 1F, G). These cells produced small outward currents at positive

membrane potentials, but these currents may at least in part be generated by PAC channels. Furthermore, even the application of 30 mM HOAc at pH 5.0 only induced small heat-evoked currents on hTRPV2-expressing cells (Appendix Fig. 1H). CBD is one of few TRPV2-agonists that robustly activates hTRPV2 (Qin et al, 2008). When we examined CBD-induced currents of hTRPV2, an HOAc-induced potentiation was only observed following washout of HOAc (Appendix Fig. 1I,J). These data show that HOAc can activate and sensitize rodent TRPV2 orthologues, but also that hTRPV2 seems to be both inhibited and activated by at least HOAc. We also examined whether activation of TRPV2 is an effect specific for HOAc, or if it applies to weak acids in general. Indeed, the weak acids carbon dioxide, lactic acid, propionic acid and formic acid activated and sensitized rTRPV2 (Fig. EV2A–O). In agreement with a recent study on TRPV2 (Catalina-Hernandez et al, 2024), we also observed an activation and sensitization of rTRPV2 by benzoic acid (Fig EV2P–S). However, the non-deprotonatable β-hydroxybutyric acid failed to activate rTRPV2 (Appendix Fig. 1K,L).

Having established that weak acids activate and sensitize rTRPV2, we next aimed to study the activation mechanism in more detail. Activation of TRPV3 and TRPA1 by weak acids seems to be mediated by an intracellular accumulation of protons driven by permeation of the undissociated form of the acid through the cell membrane (Cao et al, 2012; Wang et al, 2011). In order to examine if this property also applies to rTRPV2, we analyzed the pH-dependency (pH 7.4, 6.0, or 5.0) for activation of rTRPV2 by 10, 20, and 30 mM HOAc. Given that HOAc has a $pK_a$-value of 4.76, this experimental approach results in an increasing amount of intracellular entry of HOAc at lower pH values and high concentrations. To standardize experimental conditions between each experimental group, peak current amplitudes were determined after 3 min application of the test solutions (Fig. 2A). Indeed, amplitudes of HOAc-induced inward currents increased with higher concentrations of HOAc and with lower pH values (Fig. 2B, $n = 5–11$ for each condition). Using the pH-indicator BCECF, we found that application of 10, 20, or 30 mM HOAc at pH 7.4, 6.0, or 5.0 on HEK293T cells resulted in a reduction of the intracellular pH value that depended on both concentration of HOAc and the pH value (Fig. 2C). As HOAc at pH 7.4 both failed to activate TRPV2 and to induce a substantial intracellular acidosis, it seems likely that membrane permeability of HOAc and the resulting intracellular acidosis are required for activation of TRPV2. Accordingly, channel activation was evoked by application of 30 mM HOAc at pH 5.0 in the cell-attached configuration (Fig. 2D, $n = 5$). In all, 10 mM HOAc at pH 5.0 also induced a strong potentiation of CBD-induced membrane currents in this configuration (Fig. 2E,F, $n = 6$). Overall, 30 mM HOAc at pH 5.0 also induced channel activation in cell-free inside-out multi-channel recordings at −60 mV (Fig. 2G, $n = 5$). When single-channel recordings were performed on inside-out patches at +60 mV, the application of 30 mM HOAc at pH 5.0 induced robust channel openings with an amplitude of $4.3 \pm 0.1$ pA and a single-channel conductance of $71.6 \pm 5.2$ pS (Fig. 2H–K, $n = 6$). All patches displaying sensitivity to HOAc also generated single-channel openings following the application 500 μM 2-APB, confirming expression of TRPV2 (Fig. 2H; Appendix Fig. 2A,B). Of note, the single-channel amplitudes of currents induced by 500 μM 2-APB were larger than those induced by HOAc (Appendix Fig. 2B, $8.1 \pm 0.1$ pA). Moreover, experiments with 100 μM 2-APB or 10 μM CBD resulted in single-channel currents with amplitudes around

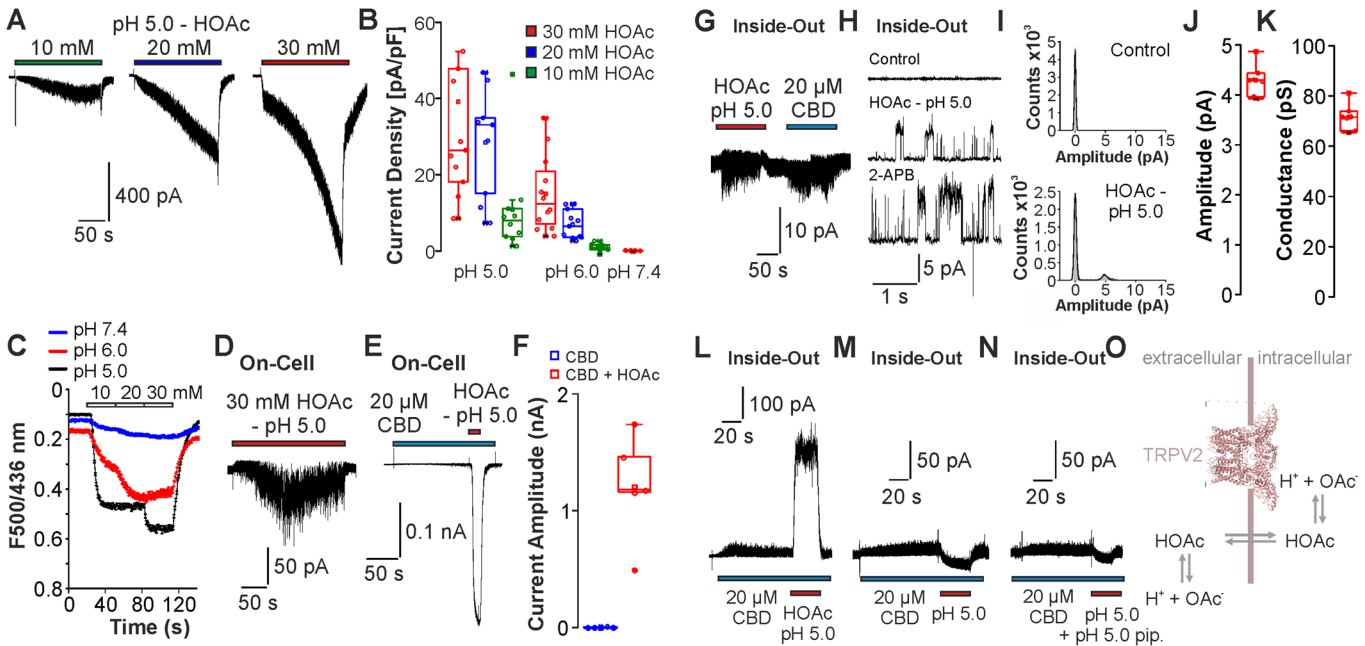

**Figure 2. Properties of weak acid-sensitivity of rTRPV2.**

(A) Typical original current traces displaying concentration-dependent (10, 20, and 30 mM at pH 5.0) HOAc-induced inward currents in rTRPV2-expressing cells held at −60 mV. (B) Box diagrams with dot plots displaying the current densities of HOAc-induced inward currents obtained at different conditions; 10, 20, and 30 mM HOAc at pH 5.0 or pH 6.0. At pH 7.4, only 30 mM HOAc was tested. Only one test solution was applied to each cell tested ($n = 5$–12). (C) BCECF-based ratiometric imaging on non-transfected HEK293T cells treated with 10, 20, or 30 mM HOAc titrated to pH 7.4, 6.0, or 5.0. A decrease in fluorescence ratio correlates with intracellular acidosis. (D) Representative on-cell recording demonstrating activation of membrane currents induced by 30 mM HOAc at pH 5.0 in a rTRPV2-expressing cell. (E) Typical on-cell recording on a rTRPV2-expressing cell demonstrating a potentiation effect of 10 mM HOAc at pH 5.0 on membrane currents induced by 20 µM CBD. (F) Box diagram with dot plots displaying the normalized amplitudes of membrane currents induced by CBD or CBD + 10 mM HOAc at pH 5.0 in cell-attached recordings ($n = 5$). (G) 30 mM HOAc at pH 5.0 induced reversible currents on inside-out multi-channel recordings on membrane patches containing rTRPV2. (H) Single-channel recordings on rTRPV2 performed on inside-out patches held at +60 mV. Channel openings were induced by 30 mM HOAc at pH 5.0 as well as by 500 µM 2-APB. (I) Amplitude histograms for control solution and 30 mM HOAc at pH 5.0, both calculated from 5-s sections of the full trace. (J, K) Box diagrams with dot plots displaying the amplitudes (J) and conductance (K) of single-channel openings induced by HOAc ($n = 6$). (L) Inside-out multi-channel recordings on membrane patches containing rTRPV2 examined at +60 mV. CBD-induced membrane currents are potentiated by 10 mM HOAc at pH 5.0 in rTRPV2-containing patches. (M, N) Both co-application of CBD and pH 5.0 from cytosolic side (M) and the combined intracellular and extracellular acidification (N) induced an inhibition of CBD-induced currents. (O) Cartoon showing how weak acids are able to induce intracellular acidosis and to interact with intracellular residues of TRPV2. (G, L) The box denotes the 50th percentile (median) as well as the 25th and 75th percentile. The whiskers mark the 5th and 95 percentiles. Data points beyond the whiskers are outliers. (B, F, J, K) The box denotes the 50th percentile (median) as well as the 25th and 75th percentile. The whiskers mark the 5th and 95 percentiles. Data points beyond the whiskers are outliers. Source data are available online for this figure.

5–6 pA (Appendix Fig. 2C–F). These data suggest that gating properties of rTRPV2 vary between different agonists, and that they are concentration-dependent. In agreement with previous reports that cytosolic protons do not activate TRPV2 (Cao et al, 2012; Gao et al, 2016), application of pH 5.0 titrated with HCl to the cytosolic side did not induce membrane currents in multi-channel inside-out recordings (Appendix Fig. 2G, $n = 5$). Application of pH 5.0 on both sides of the membrane as well as 30 mM potassium acetate at pH 7.4 also failed to activate currents in TRPV2-expressing inside-out patches (Appendix Fig. 2H,I, $n = 5$ and 4). The potentiation of CBD-induced currents by HOAc observed in whole cell as well as cell-attached experiments was very strong, probably similar to the degree of the supra-additive potentiation we and other groups recently described for the combination of CBD and 2-APB on TRPV2 (Gochman et al, 2023; Pumroy et al, 2022). This HOAc-induced potentiation of CBD-induced currents was not observed at pH 7.4 in the whole-cell recordings (Appendix Fig. 2J,K, $n = 6$), again indicating that intracellular mechanisms are involved in HOAc-induced gating of TRPV2. Considering that the

combination of CBD and HOAc gives rise to a much stronger channel activation as compared to HOAc alone, we examined the effects of HOAc or protons on CBD-induced membrane currents in cell-free inside-out patches. As expected, 10 mM HOAc at pH 5.0 induced a strong potentiation of CBD-induced currents in inside-out recordings (Fig. 2L). In contrast, an inhibition rather than a potentiation of CBD-induced currents was observed upon application of pH 5.0 titrated with HCl on the cytosolic side or with the combined acidification of the cytosolic and extracellular solutions (Fig. 2M,N, $n = 5$ each). Taken together, these data indicate that the effect of HOAc on TRPV2 cannot be reproduced by application of protons or acetate. Thus, the mechanism for weak acid-induced activation of TRPV2 seems to be more complex than a sole binding of protons on specific intracellular or extracellular binding sites (Fig. 2O).

TRPV2 was reported to interact with the actin cytoskeleton, and that this interaction renders TRPV2 mechano-sensitive (Sugio et al, 2017). Even though HOAc was found to gate TRPV2 in cell-free inside-out patches, we examined if weak acid-sensitivity

is modulation following disruption of the cytoskeleton. Application of cytochalasin-D and latranculin A (inhibitors of actin polymerization) or nocodazole (a microtubule inhibitor) did not reduce HOAc-induced inward currents in TRPV2-expressing cells (Appendix Fig. 2L–N). These data suggest that weak acid-sensitivity of TRPV2 does not depend on the cytoskeleton.

Even though our functional data did not define a clear role of extracellular or intracellular protons as important mediators of weak acid-sensitivity of TRPV2, we performed two independent approaches with the aim to identify proton modulation sites of TRPV2 being required for HOAc-induced activation and sensitization of rTRPV2. Similar to the approach that allowed for the identification of extracellular proton modulation sites of TRPV1 (Jordt et al, 2000), we first probed the outer pore domain of rTRPV2 by performing an unbiased mutational screen. All targeted residues were conservatively replaced by similar amino acids lacking a titratable moiety. Some residues (Lys566, Glu569, Asp570, Glu577, and Glu586) were also more severely mutated to alanine. Among the thirteen conservative mutants, only two (rTRPV2-E561Q and -K602Q) displayed obvious deficits for both HOAc-induced inward currents and HOAc-induced sensitization of heat-evoked currents (Fig. EV3A,B). The general functionality of all mutants was examined by testing their responses to a high concentration (1 mM) of 2-APB. Only rTRPV2-K602Q displayed markedly reduced 2-APB-induced responses (Fig. EV3C–E). Therefore, the almost complete loss of HOAc-sensitivity observed for rTRPV2-K602Q may be due to a general loss of function. Surprisingly, the alanine mutants displayed a clear gain-of-function property for activation by HOAc (Fig. EV3F–H). However, this property was not observed for potentiation of heat-induced currents or for activation by 2-APB (Fig. EV3I,J). Of note, similar effects were previously observed on pore-mutants in TRPV1 (Jordt et al, 2000). Together, this unbiased mutational screen did not deliver conclusive evidence for or against an important role of proton modulation sites in outer pore domain for weak acid-sensitivity of rTRPV2.

We next employed cryo-EM to examine the structural effects of adding 30 mM HOAc at pH 5.0 to rTRPV2. We first obtained a higher resolution apo rTRPV2 in nanodisc dataset at pH 8, yielding the same classes observed previously: TRPV2$_{Apo1}$ at 2.85 Å and TRPV2$_{Apo2}$ at 2.68 Å, as well as an additional state, TRPV2$_{Apo3}$ at 3.0 Å, similar to the CBD-bound rTRPV2$_{CBD2}$ (PDB 6U88) or the apo rabbit TRPV2 (PDB 5AN8) states (Appendix Table 1; Appendix Figs. 3–5) (Huynh et al, 2016; Pumroy et al, 2019, 2022; Zubcevic et al, 2019). The overall conformations of TRPV2$_{Apo1}$ and TRPV2$_{Apo2}$ are quite similar—closed at the lower gate and gated by Met645, N-terminus in a helical conformation, His651 constricting the bottom of the pore (Figs. 3A,B and EV4A,B,D). The major difference between these states is the status of the selectivity filter, which is tightly closed in TRPV2$_{Apo1}$ and opened wide in TRPV2$_{Apo2}$ (Fig. EV4). While the pore of TRPV2$_{Apo3}$ is otherwise very similar to that of TRPV2$_{Apo1}$, His651 is rotated out of the ion permeation path (Fig. EV4E,F). This state also features a downward rotation of the ankyrin repeat domains (ARD) and the conformational change of the C-terminus from a helix inside the ARD skirt to a loop which wraps around to interact with the N-terminal end of the ARDs (Fig. EV4G–I). Work on TRPV3 suggests that the loop conformation of the C-terminus indicates an inactivated or desensitized state, while the helix

conformation is required for an activated or sensitized state (Zubcevic et al, 2018). Therefore, we propose that TRPV2$_{Apo1}$ and TRPV2$_{Apo2}$ are in pre-open states and TRPV2$_{Apo3}$ is a desensitized channel.

We then prepared grids of rTRPV2 briefly exposed (<1 min) to 30 mM sodium acetate with a final pH of 5.0. This dataset yielded a single state, TRPV2$_{HOAc}$ at 3.14 Å, which featured an open selectivity filter and a helical C-terminus (Appendix Table 1; Fig. EV4C,F,J; Appendix Figs. 5 and 6), resembling TRPV2$_{Apo2}$. Beyond the shift in equilibrium to favor the open selectivity filter, there are other notable changes in TRPV2$_{HOAc}$. First, we examined the extracellular surface of the channel. Our initial mutagenesis data suggested two residues, Glu561 and Lys602 as possible proton modulation sites. We noticed that the voltage sensing-like domain (VSLD) of TRPV2$_{HOAc}$ shifts compared to TRPV2$_{Apo1}$ and TRPV2$_{Apo2}$ (Fig. EV5A,B). In addition, we see an inward movement of S5 (Fig. EV5A,B) and a corresponding change at the bottom of S6 compared to TRPV2$_{Apo2}$ (Fig. EV4A,B). While these changes did not lead to the opening of the lower gate at Met645, these movements coincide with a 180° rotation of Glu495 at the top of S4, away from the membrane interface towards S5 and swapping positions with Arg617 (Fig. 3A–C). We used PROPKA3 to determine that Glu495 and Glu561 are likely to be protonated at pH 5.0 in TRPV2$_{HOAc}$ (Appendix Table 2), which could allow for the formation of a carboxyl-carboxylate interaction between the two glutamic acids when in close proximity. In order to determine if Glu495 and Glu561 are relevant for activation by weak acids, we examined the mutants rTRPV2-E495Q (Fig. 3D, $n = 9$) and rTRPV2-E561Q (Fig. 3E $n = 8$). Indeed, both mutants displayed significantly reduced HOAc-induced inward currents (Fig. 3F, $n = 9$, one-way ANOVA, Bonferroni correction, DF = 2, $F$ value: 8.30592). HOAc-induced potentiation of heat-evoked currents was also reduced for both mutants (Fig. 3G–I, $n = 5$ for both mutants, one-way ANOVA, DF = 2, $F$ value: 36.28001). Of note, these residues are highly conserved in mammalian TRPV2, but not across the TRPV family (Appendix Fig. 3). We then ran molecular dynamic (MD) simulations starting from TRPV2$_{Apo1}$ both with and without protonation of key residues, including Glu495 and Glu561. Consistent with the cryo-EM predictions, during MD simulations with the protonated residues (simulating pH 5) the Glu561–Arg617 contact is present in 67.15% of frames and the Glu561–Glu495 contact is present in 16.57% of frames compared to 87.07% for Glu561–Arg617 and 0.003% for Glu561–Glu495 in the MD simulation with the unprotonated residues (simulating pH 7) (Fig. 3J–L). These data suggest that Glu495 and Glu561 may be relevant, but not required for activation of TRPV2 by weak acids. The structures also suggest a reason for the loss of function of the K602Q mutant: in the open selectivity filter conformations, TRPV2$_{Apo2}$ and TRPV2$_{HOAc}$, Lys602 is engaged in a cation-π interaction with Phe612, which is likely essential for stabilizing the open conformation of the selectivity filter (Fig. EV5C–E).

We next examined the intracellular surface of the channel and observed that His651 is rotated out of the path of the central channel along with the shifts in S5 and S6 (Fig. EV4C). We also observed a major disruption at the S4–S5 linker, where His521 rotates away from S5 (Fig. 4A–C). His651 is unlikely to be the primary source of weak acid-sensitivity as this residue is only found in rat TRPV2 (Appendix Fig. 3). The MD simulations validate this notion as the protonation of His651 does not significantly alter its

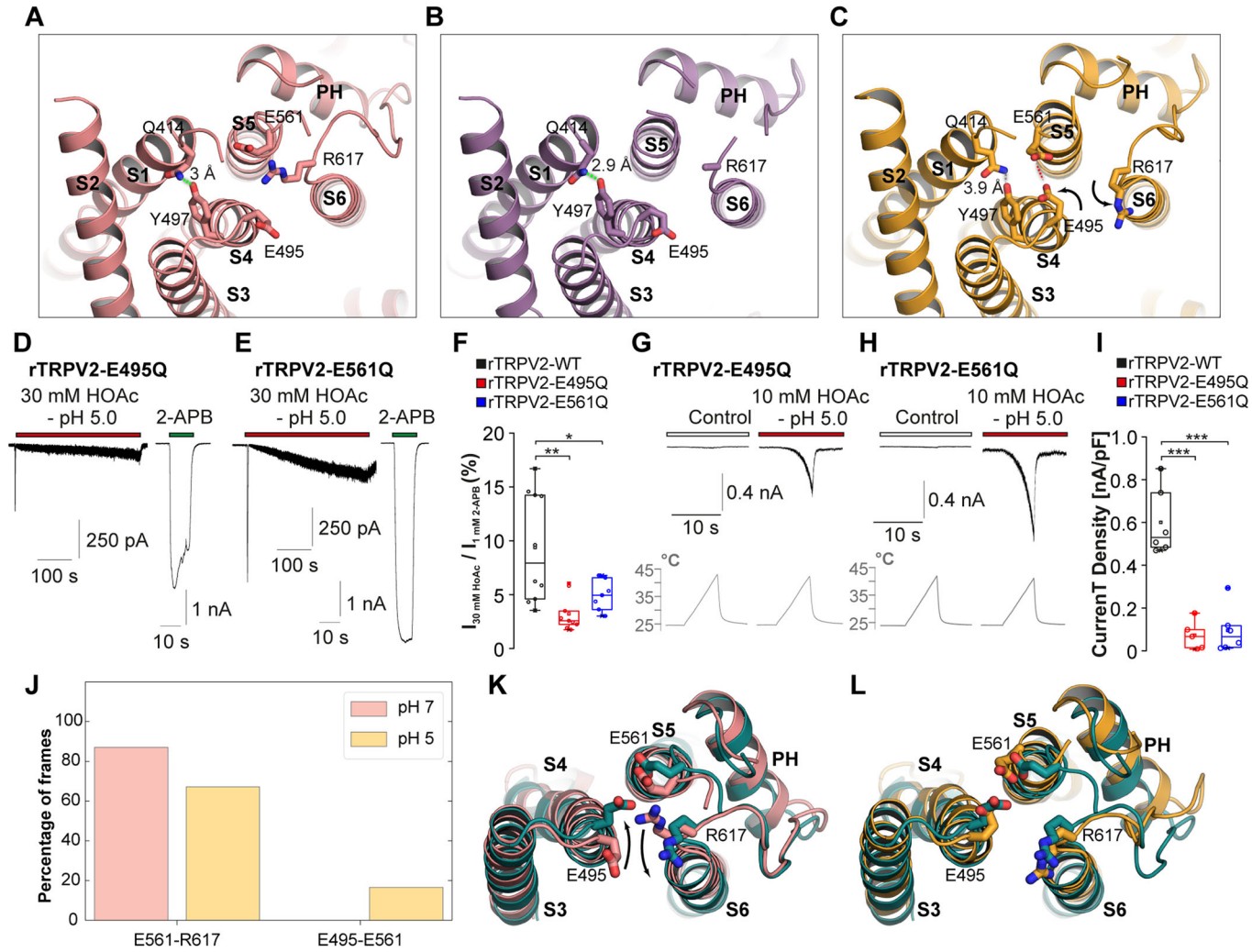

**Figure 3. Identification and validation of extracellular proton modulation sites.**

(A–C) View of the top of the VSLD of TRPV2$_{Apo1}$ (A), TRPV2$_{Apo2}$ (B), and TRPV2$_{HOAc}$ (C). A hydrogen bond between Gln414 and Tyr497 in TRPV2$_{Apo1}$ and TRPV2$_{Apo2}$ is indicated by a green dashed line, the broken hydrogen bond in TRPV2$_{HOAc}$ is indicated by a gray dashed line. A possible carboxyl-carboxyl interaction between Glu495 and Glu561 is indicated by a red dashed line. (D, E) Representative current traces from HEK293T cells expressing rTRPV2-E495Q (D) or rTRPV2-E561Q (F). While 30 mM HOAc at pH 5.0 evoked small currents on this mutant, 2-APB induced large currents. (F) Box diagrams with dot plots displaying the relative magnitudes of HOAc-evoked inward currents normalized with the responses evoked by 1 mM 2-APB. Data are displayed for rTRPV2-WT ($n = 10$), -E495Q ($n = 9$) and -E561Q ($n = 7$). (G, H) Heat-evoked currents in cells expressing rTRPV2-E495Q (G) and rTRPV2-E561Q (H) 10 mM HOAc at pH 5.0 induced robust but comparably small heat-evoked currents. (I) Box diagrams with dot plots displaying the current densities of heat-evoked currents provoked by 10 mM HOAc at pH 5.0 in cells expressing rTRPV2-WT ($n = 6$), -E495Q ($n = 5$), and -E561Q ($n = 6$). (J) Stability of the Glu495-Glu561 interaction (orange) or Arg617-Glu561 salt-bridge (salmon) within the final 120 ns of MD simulations initiated from TRPV2$_{Apo1}$ under pH 7 and pH 5 protonation conditions. The two residues were assumed to be in contact if the distance between any of their non-hydrogen atoms was <3.3 Å. (K, L) Snapshot illustrating the swapping of interactions within the extracellular surface of the channel. When initiated from TRPV2$_{Apo1}$ (salmon), in MD simulations under protonating conditions (dark teal) Arg617 and Glu495 swap positions (K), which leads to a configuration similar to TRPV2$_{HOAc}$ (L, orange). *$P < 0.05$, **$P < 0.01$, and ***$P < 0.001$. One-way ANOVA, Bonferroni post hoc test. (F, I) The box denotes the 50th percentile (median) as well as the 25th and 75th percentile. The whiskers mark the 5th and 95 percentiles. Data points beyond the whiskers are outliers. Source data are available online for this figure.

motility and orientation relative to the central channel pore (Appendix Fig. 7). His521 is highly conserved in most TRPV2 orthologues, but it is not observed in this position in other TRPV channels (Appendix Fig. 3). His521 is found at the N-terminal end of the S4–S5 linker, exposed to the lipid interface. In all three apo TRPV2 structures, His521 forms a cation-π interaction with Arg539 from the adjacent TRPV2 monomer (Fig. 4A,B). In TRPV2$_{HOAc}$, His521 is rotated away from Arg539, possibly due to protonation (Fig. 4C). The combined effect of protonation and

rotation exposes a positively charged pocket comprised of His521, Arg539, and Arg535. In an intriguing parallel, we recently observed that the activator 2-APB bound in the same pocket (Pumroy et al, 2022), inducing an outward rotation of His521 similar to what we observe in TRPV2$_{HOAc}$ (Fig. 4D). PROPKA3 predicted that His521 would be protonated in the TRPV2$_{HOAc}$ structure (Appendix Table 2). We also observed that protonation of His521 caused a splaying of the S4–S5 linker and S5 helices in a subset of subunit interfaces (Fig. 4E,F). This splaying was similar in extent to that

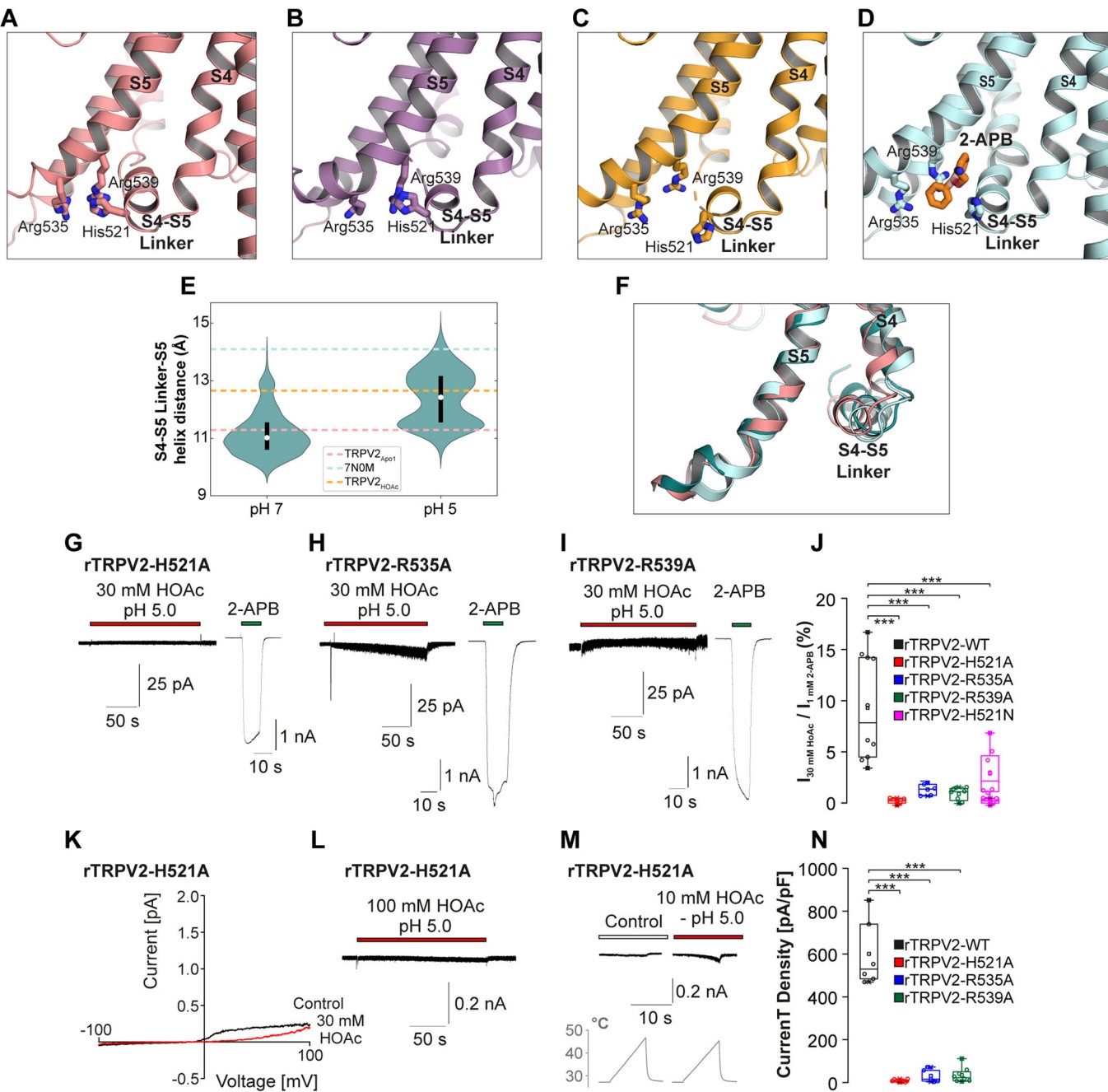

observed in the rTRPV2 inactivated 2-APB-bound structure (PDB ID: 7N0M, Fig. 4E,F). In contrast, the helices remain closer together at a distance observed in TRPV2$_{Apo1}$ with a neutral His521 (Fig. 4F). We next performed patch clamp experiments on the mutant rTRPV2-H521A as the putative proton modulation site, but also rTRPV2-R535A and rTRPV2-R539A as critical residues of this positively charged pocket. When challenged with 30 mM HOAc at pH 5.0, rTRPV2-H521A failed to generate inward currents (Fig. 4G, $n = 6$). Only minimal HOAc-induced currents were observed in cells expressing rTRPV2-R535A (Fig. 4H, $n = 6$) and rTRPV2-R539A (Fig. 4I, $n = 8$). All three mutants are functional as they generated large inward currents following application of 1 mM 2-APB (Appendix Fig. 8A), and we previously demonstrated that

they are activated by heat (Pumroy et al, 2022). When normalizing the amplitudes of HOAc-induced currents with currents induced by 1 mM 2-APB, all three mutants displayed an almost complete loss in HOAc-sensitivity (Fig. 4J, one-way ANOVA, Bonferroni correction, DF = 4, F value: 15,57216). A replacement of His521 with an asparagine instead of an alanine also showed minimal HOAc-induced currents (rTRPV2-H521N, Fig. 4J, $n = 8$). TRPV2-H521A also completely failed to generate membrane currents when exposed to 30 mM or even 100 mM HOAc at pH 5.0 (Fig. 4K,L, $n = 5$ for each experiment). Furthermore, 10 mM HOAc at pH 5.0 induced negligible heat-evoked currents on cells expressing rTRPV2-H521A, rTRPV2-R535A or rTRPV2-R539A (Fig. 4M,N, $n = 6–8$, one-way ANOVA, Bonferroni post hoc test). Finally,

**Figure 4.   Identification and validation of intracellular proton modulation site.**

(A–D) View of the putative intracellular proton modulation site in TRPV2$_{Apo1}$ (A), TRPV2$_{Apo2}$ (B), TRPV2$_{HOAc}$ (C) and 2-APB-bound TRPV2 (PDB 7N0M, (D). (E) Splaying of the S4–S5 linker and S5 helices calculated as the distance between the center-of-mass of the C$\alpha$ atoms of residues 519–525 (S4–S5 linker) residue and the center-of-mass of the C$\alpha$ atoms of residues 534–540 (S5). Violin plots contain data calculated from the final 120 ns of trajectory ($n = 32,000$; 2000 frames each from 4 subunits and 4 simulation replicas). The median values are represented as white dots and the interquartile range is shown as a line. The same distances within TRPV2$_{Apo1}$, TRPV2$_{HOAc}$ and the 2-APB-bound TRPV2 (PDB 7N0M) structure are shown as dotted lines. (F) Snapshot illustrating the splaying of the S4–S5 linker and S5 helices. When initiated from TRPV2$_{Apo1}$ (salmon), in MD simulations under protonating conditions (dark teal) the distance between the helices increased to that observed in the 2-APB-bound structure (PDB 7N0M, cyan). (G–I) Representative current traces from HEK293T cells expressing rTRPV2-H521A (E), rTRPV2-R535A (F), and rTRPV2-R539A (G). In all three mutants, 30 mM HOAc at pH 5.0 evoked no or only very small currents. Functionality and expression were validated by the application of 1 mM 2-APB. (J) Box diagrams with dot plots displaying the relative magnitudes of HOAc-evoked inward currents normalized with the responses evoked by 1 mM 2-APB. Data are displayed for rTRPV2-WT ($n = 10$), -H521A ($n = 6$), -H521N ($n = 8$), -R535A ($n = 6$), and -R539A ($n = 8$). (K) Current trace displaying membrane currents during a voltage ramp from $-100$ to $100$ mV in a cell expressing rTRPV2-H521A. Note that 30 mM HOAc at pH 5.0 inhibited the basal current without evoking any response. (L) Example of a current trace displaying that even 100 mM HOAc at pH 5.0 fails to activate rTRPV2-H521A. (M) Heat-evoked currents in cell expressing rTRPV2-H521A. Note that 10 mM HOAc at pH 5.0 only provokes a minimal heat-evoked current. (N) Box diagrams with dot plots displaying the current densities of heat-evoked currents provoked by 10 mM HOAc at pH 5.0 in cells expressing rTRPV2-WT ($n = 6$), -H521A ($n = 8$), -R535A ($n = 5$), and -R539A ($n = 8$). The symbols between Fig. 5J, N apply for both panels. ***$P < 0.001$. One-way ANOVA, Bonferroni post hoc test. (J, N) The box denotes the 50th percentile (median) as well as the 25th and 75th percentile. The whiskers mark the 5th and 95 percentiles. Data points beyond the whiskers are outliers. Source data are available online for this figure.

rTRPV2-H521A displayed a reduced potentiation of CBD-induced currents by HOAc (Appendix Fig. 8B,C). Together, these data clearly indicate that the positively charged pocket comprised of His521, Arg539 and Arg535 is critical for weak acid activation and sensitization of TRPV2. As the exchange of H521 resulted in an almost complete loss of weak acid-sensitivity, it seems justified to argue that this residue is a critical proton modulation site.

What is the physiological relevance of weak acid-sensitivity of TRPV2? This study does not include more intact in vivo or ex vivo experimental preparations required to address this question, but we examined the effects of weak acids on the properties of TRPV2 that seem to be of physiological relevance. TRPV2 displays a high permeability for extracellular Ca$^{2+}$ (Caterina et al, 1999), and an increase of cytosolic Ca$^{2+}$ is decisive for several processes depending on TRPV2 (Conde et al, 2021; Guo et al, 2022; Maksoud et al, 2019; Shoji et al, 2023). We examined if activation of TRPV2 by weak acids is associated with a Ca$^{2+}$-influx. To our surprise, HEK293T cells expressing rTRPV2 did not exhibit a robust elevation of intracellular Ca$^{2+}$ following the application of weak acids. HOAc-induced effects in rTRPV2-expressing cells ($n = 237$) were only marginally larger than the effects observed in non-transfected cells ($n = 422$) (Fig. 5A). A similar effect was obtained with 100 mM propionic acid at pH 6.0 (Fig. 5B, $n = 153$ transfected and 351 non-transfected). The patch clamp data obtained on TRPV2 in this study include experiments performed with nominal Ca$^{2+}$-free solutions in order to avoid a Ca$^{2+}$-dependent desensitization (Mercado et al, 2010). Therefore, these Ca$^{2+}$ imaging data indicate that extracellular calcium inhibits weak acid-induced activation of TRPV2. Indeed, patch clamp experiments with 2 or 10 mM Ca$^{2+}$ in the extracellular solution revealed that HOAc-induced inward currents through rTRPV2 were significantly reduced (Fig. 5C–E, $n = 6$ and 8, one-way ANOVA with Bonferroni correction, DF = 2, $F$ value: 15.88024). Furthermore, inward currents induced by HOAc in Ca$^{2+}$-free extracellular solution were strongly inhibited when 2 mM Ca$^{2+}$ was co-applied (Fig. 5E, $n = 7$). A similar Ca$^{2+}$-induced inhibition was previously described for TRPV3 (Xiao et al, 2008). Therefore, we also explored if similar effects of Ca$^{2+}$ are observed on inward currents induced by CBD and 2-APB, which both evoke a strong TRPV2-mediated Ca$^{2+}$-influx when applied alone (Neeper et al, 2007; Qin et al, 2008). Different from HOAc-induced currents, inward currents induced by both 20 $\mu$M CBD or 500 $\mu$M 2-APB in Ca$^{2+}$-free solution exhibited a transient rapid current increase upon application of 2 mM Ca$^{2+}$ (Appendix Fig. 9A,B). These data indeed indicate that a strong inhibition of TRPV2 by extracellular Ca$^{2+}$ only applies for weak acid-induced activation. In TRPV3, both calcium- and proton-induced inhibition are dictated by the pore residue Asp641 (Wang et al, 2021; Xiao et al, 2008). The equivalent residue in TRPV2 is Glu609, i.e., one of the pore residues that we already examined as a possible proton modulation site (Fig. EV4). We found that Ca$^{2+}$-induced inhibition of HOAc-induced currents was reduced for rTRPV2-E609Q (Fig. 5F,G, $n = 6$, $P < 0.001$ unpaired $t$ test). However, the biphasic effect of Ca$^{2+}$ on 2-APB-induced currents was maintained on rTRPV2-E609Q (Appendix Fig. 9C). We also observed that application of pH 5.0 titrated with HCl induced a partial inhibition of CBD-induced currents generated by wild-type rTRPV2 (Appendix Fig. 9D). This proton-induced inhibition was significantly increased on rTRPV2-E609Q (Appendix Fig. 9D–F, $P < 0.01$, unpaired $t$ test). These data show that the pore residue Glu609 in rTRPV2 is an important determinant for the inhibition of weak acid-induced currents by extracellular Ca$^{2+}$. In contrast to Asp641 in TRPV3 however, Glu609 does not seem to dictate inhibition of TRPV2 by Ca$^{2+}$ or protons in general.

The failure of weak acids to induce a robust TRPV2-mediated Ca$^{2+}$-influx raised the critical question as to whether weak acids are likely to be relevant endogenous modulators of TRPV2 expressed in the cell membrane or not. We examined if an HOAc-induced potentiation of TRPV2 activation induced by other agonists is observed in the presence of extracellular Ca$^{2+}$. 10 mM HOAc at pH 5.0 induced a strong potentiation of both heat-induced currents (Fig. 5H, $n = 6$) and CBD-induced currents (Fig. 5I,J, $n = 8$) in the presence of 2 mM Ca$^{2+}$. Furthermore, Ca$^{2+}$-imaging experiments demonstrated that 10 mM HOAc at pH 5.0 strongly potentiates CBD-induced Ca$^{2+}$-influx in rTRPV2-expressing cells, but not in non-transfected HEK293 cells (Fig. 5K, $n = 597$ and 388, respectively). These data demonstrate that weak acids can sensitize TRPV2 at physiological concentrations of extracellular calcium.

We also asked if the inhibition by extracellular Ca$^{2+}$ is associated with a lower permeability of rTRPV2 to Ca$^{2+}$ when it is activated by HOAc as compared to 2-APB or CBD. Of note, the frequently cited

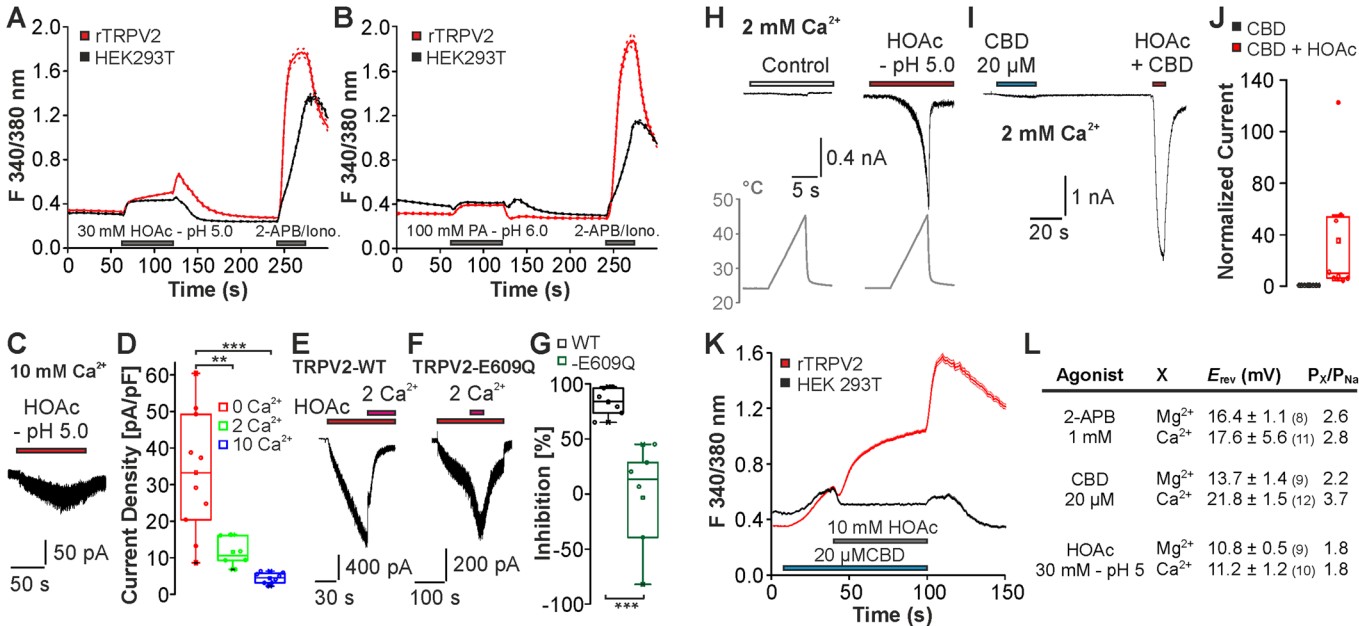

**Figure 5. Extracellular Ca²⁺ inhibits HOAc-induced activation of rTRPV2.**

(A, B) Mean increase in F340/380 nm ratio evoked by 30 mM HOAC at pH 5.0 (A) and 100 mM propionic acid at pH 6.0 (B) in naive HEK293T cells or in HEK293T cells expressing rTRPV2. In total, 1 mM 2-APB was applied at the end of the experiment to verify expression rTRPV2. Ionomycin was applied on non-transfected cells in order to verify calcium influx in cells. (C) Typical whole-cell recordings displaying HOAc-evoked inward currents in rTRPV2-expressing cells in the presence of 10 mM Ca²⁺ in the extracellular solution. (D) Box diagrams with dot plots displaying current densities induced by 30 mM HOAc—pH 5.0 in nominal Ca²⁺-free solution ($n = 10$), 2 mM Ca²⁺ ($n = 6$), or 10 mM Ca²⁺ ($n = 8$). (E, F) Original current traces on rTRPV2-WT (E) or the mutant rTRPV2-E609Q (F) displaying inhibition of HOAc-induced currents by 2 mM Ca²⁺. Inward currents were induced by 30 mM HOAc at pH 5.0 in Ca²⁺-free solution. 2 mM Ca²⁺ was co-applied with 30 mM HOAc at pH 5.0. Note that HOAc-evoked currents of rTRPV2-E609Q developed very slowly. (G) Box diagram with dot plots displaying Ca²⁺-induced inhibition of HOAc-evoked currents of rTRPV2-WT ($n = 7$) or rTRPV2-E609Q ($n = 6$). (H) Trace displaying HOAc-induced sensitization of rTRPV2 to heat in presence of 2 mM Ca²⁺. Solutions were applied as "heat ramps" from room temperature to 40–45 °C. (I) Current traces on rTRPV2-expressing cells with two consecutive applications of 20 μM CBD in presence of 2 mM Ca²⁺. For the 2nd application, CBD was combined with 10 mM HOAC at pH 5.0. (J) Box diagram with dot plots displaying normalized current amplitudes of currents in induced by CBD (set as 1) or CBD + HOAc ($n = 8$). (K) Mean increase in F340/380 nm ratio evoked by 20 μM CBD followed by CBD + 10 mM HOAC at pH 5.0 in naive HEK293 cells or in HEK293 cells expressing rTRPV2. In total, 1 mM 2-APB was applied at the end of the experiment verify expression rTRPV2. Ionomycin was applied on non-transfected cells in order to verify calcium influx in cells. (L) Reversial potentials and relative cationic permeabilities of membrane currents induced by 2-APB, CBD or HOAc in cells expressing rTRPV2. Values are means ± SEM and the numbers of investigated cells are given in brackets. **$P < 0.01$ and ***$P < 0.001$. One-way ANOVA, Bonferroni post hoc test. (D, G, J) The box denotes the 50th percentile (median) as well as the 25th and 75th percentile. The whiskers mark the 5th and 95 percentiles. Data points beyond the whiskers are outliers. Source data are available online for this figure.

high Ca²⁺-permeability of rTRPV2 was only determined following activation by heat exceeding 50 °C (Caterina et al, 1999). As is demonstrated in Fig. 5L, we determined a high permeability for both Ca²⁺ and Mg²⁺ when rTRPV2 was activated by 2-APB and CBD. As postulated, activation by HOAc was associated with lower permeabilities for both Ca²⁺ and Mg²⁺. Thus, both divalent ion permeabilities and single-channel conductances of TRPV2 seem to vary between agonists.

Both ASIC1a and PAC channels, but also TRPV3 were demonstrated to mediate cytotoxicity induced by acids (Cao et al, 2012; Sherwood et al, 2011; Ullrich et al, 2019; Yang et al, 2019). Therefore, we finally asked if the expression of rTRPV2 in HEK293 cells reduces cell vitality following incubation with HOAc. While incubation in pH 5.0 titrated with HCl for 2 h did not result in a substantial cytotoxicity (Fig. 6E), treatment with 10 mM and 30 mM HOAc at pH 5.0 for 2 h resulted in an increased staining with propidium iodide in both non-transfected HEK293 cells and in cells expressing rTRPV2 (Fig. 6A–E). However, an increased cell death in rTRPV2-expressing cells was only observed at 30 mM HOAc (Kruskal–Wallis with Dunn's correction, $n = 4$ repeats of 5 analyzed high power fields each).

# Discussion

There is growing evidence for a critical involvement of TRPV2 in several physiological and pathophysiological processes. This emphasizes the need for a more detailed understanding of the functional properties of this yet poorly understood ion channel, including the identification of endogenous modulators. Recent studies have made some progress on this point by describing a modulation of TRPV2 by oxidants, cholesterol and a tyrosine-dependent phosphorylation (Fricke et al, 2019; Mo et al, 2022; Su et al, 2023). In this study we describe weak acids as possible endogenous modulators of TRPV2, and our comprehensive functional and structural data give detailed insights into the mechanism accounting for weak acid-sensitivity of TRPV2.

We and other laboratories have recently identified distinct intracellular binding sites for the non-selective TRPV2-agonists 2-APB, cannabidiol and the cannabinoid C16 (Pumroy et al, 2019, 2022; Zhang et al, 2022). Although TRPV1, TRPV2 and TRPV3 have homologous structures and share ~40% sequence identity, it has become clear that 2-APB interacts with distinct

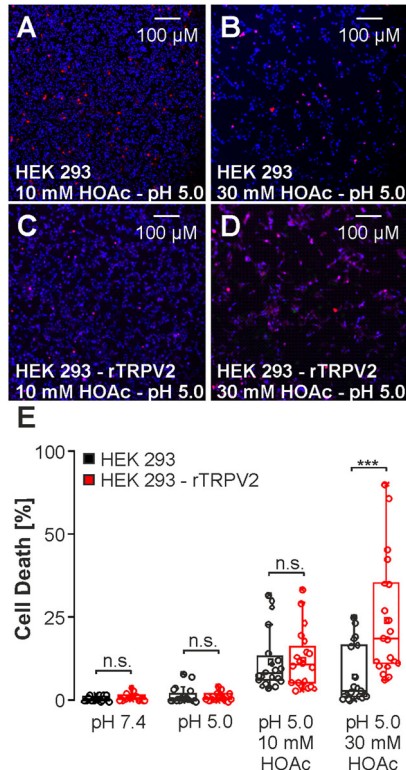

**Figure 6. Role of rTRPV2 for HOAc-induced cell death.**

(A–D) Representative figures of naive HEK293 cells (A, B) or HEK293 cells expressing rTRPV2 (C, D) following treatment with 10 or 30 mM HOAc at pH 5.0 for 2 h. Cells were double stained with propidium iodide (PI, red) and DAPI (blue). (E) Box diagrams with dot plots displaying the percentage of PI-stained cells following 2 h incubation with pH 7.4, pH 5.0 titrated with HCl and 10 or 30 mM HOAc at pH 5.0 (each condition *n* = 4 independent experiments). n.s. denotes not significant, ***$P < 0.001$. One-way ANOVA, Bonferroni post hoc test. (E) The box denotes the 50th percentile (median) as well as the 25th and 75th percentile. The whiskers mark the 5th and 95 percentiles. Data points beyond the whiskers are outliers. Source data are available online for this figure.

binding sites to activate the three ion channels (Boukalova et al, 2010; Hu et al, 2009; Pumroy et al, 2019; Singh et al, 2019). 2-APB induces a concentration-dependent intracellular acidosis when applied on HEK293 cells (Chokshi et al, 2012), possibly suggesting that 2-APB and weak acids target common binding sites to modulate TRP channels. Indeed, His426 seems to be a common intracellular activation site for both 2-APB and intracellular protons in TRPV3 (Cao et al, 2012; Hu et al, 2009). We previously determined that His521 is important for activation by 2-APB (Pumroy et al, 2022), and in this work the structure of TRPV2$_{HOAc}$ guided us to His521 as a candidate intracellular proton modulation site. Both MD simulations and the almost complete loss of weak acid-sensitivity of the rTRPV2-H521A and rTRPV2-H521N mutants perfectly confirmed this interpretation for His521. The structure of TRPV2$_{HOAc}$ also identified Glu495 and Glu561 as possible extracellular proton modulation sites, and an exchange of Glu495 and Glu561 indeed resulted in reduced sensitivities to HOAc. However, an altered HOAc-sensitivity was also determined for rTRPV2-E609Q and the alanine mutants at the outer pore of rTRPV2. Thus, the exchange of any residue in the outer pore region may result in altered functional properties, making it challenging to

define the role of distinct residues for weak acid-sensitivity. Our functional data more or less discard a simple activation mechanism only requiring binding of extracellular and/or intracellular protons to distinct residues of rTRPV2. In fact, both intracellular and extracellular application of pH 5.0 titrated with HCl even inhibited CBD-induced membrane currents. Thus, in contrast to the weak acid-sensitive channels TRPV3 and TRPA1 which are directly activated by intracellular protons (Cao et al, 2012; Gao et al, 2016; Wang et al, 2011), the mechanism accounting for weak acid-sensitivity of TRPV2 seems to be more complex.

Although HOAc-sensitivity was almost abolished in the rTRPV1-H521A mutant, further binding sites may be involved. To this end, a recent study suggested that activation of TRPV2 by probenecid and benzoic acid derivatives may be due to an interaction with the proposed binding sites of the TRPV2-inhibitor piperlongumine, including the residue Arg539 (Catalina-Hernandez et al, 2024; Conde et al, 2021). While 2-APB-sensitivity of rTRPV2 only partially depends on His521, a stronger reduction of the 2-APB-sensitivity was observed in the double mutant rTRPV2-H521A/R539K (Pumroy et al, 2022). We suggested that 2-APB is likely to interact with a positively charged pocket formed by His521, Arg535, and Arg539 (Pumroy et al, 2022), and our data may indicate that the total charge of this pocket is decisive for weak acid-sensitivity as well. Whether or not this also applies for probenecid needs to be analyzed in more detail. Although His521, Arg535 and Arg539 are conserved in most other mammalian orthologues including hTRPV2, it is intriguing that both 2-APB and weak acids almost completely fail to activate hTRPV2 (Fricke et al, 2019; Neeper et al, 2007). Our data also show that HOAc-induced potentiation of CBD-induced currents in hTRPV2-expressing cells only becomes demasked following washout of HOAc. This effect may be explained by a pore block of hTRPV2 by protons, or by HOAc itself. The residues encoding for this property of hTRPV2 are yet to be identified, and in general the function of hTRPV2 is still poorly understood. As our data leave little doubt that weak acids gate TRPV2 only when they permeate the cell membrane, it is possible that the gating mechanism involves a modification of TRPV2-independent membrane properties that result in TRPV2 sensitization and activation. HOAc and other weak acids have been shown to induce several effects when permeating through the membrane, including an increase of lipid solubility, alterations of membrane elasticity and stiffness and a dissipating of the proton motive force across the membrane (Angelova et al, 2018; Guldfeldt and Arneborg, 1998; Zhou and Raphael, 2005; Zivanov et al, 2020). It is also possible that weak acid-sensitivity of TRPV2 requires an accumulation of the intracellular anion of the weak acid. We excluded that intracellular acetate activates TRPV2, but at this point we cannot say if a simultaneous interaction of protons and anions with intracellular or extracellular residues of TRPV2 is relevant.

Another relevant finding in this study is the $Ca^{2+}$-induced inhibition of HOAc-induced activation of TRPV2. A similar property was described for TRPV3 (Wang et al, 2021; Xiao et al, 2008), and our data indicate the pore residue Glu609 dictates this property. This strong inhibition induced by extracellular calcium may give an indication of the physiological role of TRPV2 as a sensor for weak acids. TRPV2 seems to be functionally expressed in intracellular membranes such as endosomes (Cohen et al, 2015; Saito et al, 2007), or in case of the heart in the intercalated discs (Katanosaka et al, 2014). Considering that the intracellular concentration of $Ca^{2+}$ is very low (~100 nM), it is possible that the regulation of TRPV2-activity by weak acids is more relevant in intracellular membranes. Saying this, our data also clearly indicate that weak acids can strongly sensitize TRPV2 at physiological concentrations of extracellular calcium. The

physiological meaning of the acid-sensitivities of TRPV channels is generally poorly understood, and in vivo models specifically exploring acid-sensitivity are scarce. TRPV1 is thought to mediate acid-evoked pain, but aside from in vitro electrophysiological data, this assumption is probably best supported by studies using selective TRPV1-inhibitors in the acetic-acid writhing test in mice as well as in a model of acid-evoked pain in human volunteers (Caterina et al, 2000; Heber, Ciotu et al, 2020; Ikeda et al, 2001; Leffler et al, 2006). While weak acid-sensitivity of TRPV3 was suggested to account for a cytotoxic effect on keratinocytes leading to an exfoliation of the skin (Cao et al, 2012), activation of TRPA1 may explain the pungency of some weak acids (Wang et al, 2011). Our data indicate that activation of TRPV2 by weak acids can contribute to cell injury as well, a property that might be relevant in conditions associated with severe lactic or hypercapnic acidosis. The global TRPV2-knockout mouse suffers from a high perinatal lethality (Park et al, 2011), and to date selective modulators of TRPV2 are not available. Thus revealing the physiological relevance of weak acid-sensitivity of TRPV2 will require innovative strategies, and our mechanistic data offers a framework for future studies.

To conclude, we have identified weak acids as endogenous substances which are able to sensitize and even directly activate TRPV2. Our functional and structural data strongly suggest that the pocket formed by the positively charged residues His521, Arg535, and Arg539, located between S5 and the S4–S5 linker of two monomers, is likely to be an important interaction site for weak acids and further exogenous and endogenous TRPV2-ligands. This reinforces the title of the S4–S5 linker as a "gearbox" of TRP channels (Hofmann et al, 2017), and our data may facilitate the development of selective compounds targeting TRPV2.

# Methods

## Chemicals

Formic acid, lactic acid, propionic acid, benzoic acid, β-hydroxybutyric acid were purchased from Sigma-Aldrich (Taufkirchen, Germany). Acetic acid was obtained from Carls Roth GmbH+ Co KG (Karlsruhe, Germany). Acids were either dissolved or diluted in external solution and titrated to correct pH right

**Reagents and tools table**

| Reagent/resource | Reference or source | Identifier or catalog number |
|---|---|---|
| **Experimental models** | | |
| Human Embryonic Kidney 293 (HEK293) cells | ATCC | Cat# CRL-1573 |
| HEK293T cells | ATCC | Cat# CRL-3216 |
| *Saccharomyces cerevisiae* BJ5457 | ATCC | Cat# 208282 |
| ND7/23 cells | The European Collection of Authenticated Cell Cultures (ECACC) | Cat# 92090903 |
| **Recombinant DNA** | | |
| Purified rat TRPV2 | Huynh et al, 2016 https://doi.org/10.1038/ncomms11130 | N/A |
| rat TRPV2-pCDNA3.1 plasmid | Caterina et al, 1999 https://doi.org/10.1038/18906 | N/A |
| mouse TRPV2-pCDNA3.1 plasmid | Kanzaki et al, 1999 https://doi.org/10.1038/11086 | N/A |
| human TRPV2-pCDNA3.1 | This study and Fricke et al, 2019 | N/A |
| **Antibodies** | | |
| 1D4 primary antibody | Hodges et al, 1988 | N/A |
| **Oligonucleotides and sequence-based reagents** | | |
| rTRPV2 E569A | Eurogentec | N/A |
| 5'-CTGTGGAGTTGTTATCTGCAGGGGCTTTGGGACTT-3' | | |
| 5'-AAGTCCCAAAGCCCCTGCAGATAACAACTCCACAG-3' | | |
| rTRPV2-D570A | | |
| 5'-CACTGTGGAGTTGTTAGCTTCAGGGGCTTTGGG-3' | | |
| 5'-CCCAAAGCCCCTGAAGCTAACAACTCCACAGTG-3' | | |
| rTRPV2 E577A | | |
| 5'-CACCGTGGGCTGTGCCGTCACTGTGGA-3' | | |
| 5'-TCCACAGTGACGGCACAGCCCACGGTG-3' | | |
| rTRPV2 E561A | | |
| 5'-GGGACTTCGGGCCGCTCTGCTCAAGCT-3' | | |
| 5'-AGCTTGAGCAGAGCGGCCCGAAGTCCC-3' | | |
| rTRPV2 K566A | | |

| Reagent/resource | Reference or source | Identifier or catalog number |
|---|---|---|
| 5'-GTTATCTTCAGGGGCTGCGGGACTTCGGGCCTCT-3' | | |
| 5'-AGAGGCCCGAAGTCCCGCAGCCCCTGAAGATAAC-3' | | |
| rTRPV2 E586A | | |
| 5'-ATATGGAGCTGGCGCCTCCTCCTGGCC-3' | | |
| 5'-GGCCAGGAGGAGGCGCCAGCTCCATAT-3' | | |
| rTRPV2-E561Q | | |
| 5'-CTTCGGGCCTGTCTGCTCAAGCTTACTAGG-3' | | |
| 5'-CCTAGTAAGCTTGAGCAGACAGGCCCGAAG-3' | | |
| rTRPV2 K566Q | | |
| 5'-GTTATCTTCAGGGGCCTGGGGACTTCGGGCCTC-3' | | |
| 5'-GAGGCCCGAAGTCCCCAGGCCCCTGAAGATAAC-3' | | |
| rTRPV2 E569Q | | |
| 5'-CACTGTGGAGTTGTTATCCTGAGGGGCTTT<br>GGGACTTCG-3' | | |
| 5'-CGAAGTCCCAAAGCCCCTCAGGATAACAACT<br>CCACAGTG-3' | | |
| rTRPV2-D570N | | |
| 5'-CACTGTGGAGTTGTTATTTTCAGGGGCTTTGGGAC-3' | | |
| 5'-GTCCCAAAGCCCCTGAAAATAACAACTCCACAGTG-3' | | |
| rTRPV2 E577Q | | |
| 5'-CCACCGTGGGCTGCTGCGTCACTGTGGAG-3' | | |
| 5'-CTCCACAGTGACGCAGCAGCCCACGGTGG-3' | | |
| rTRPV2 E586Q | | |
| 5'-ATGGAGCTGGCTGCTCCTCCTGGCC-3' | | |
| 5'-GGCCAGGAGGAGCAGCCAGCTCCAT-3' | | |
| rTRPV2-H521N | | |
| 5'-CTGTAGATGCCTGTGTTCTGAAAGCCCCGTGTG-3' | | |
| 5'-CACACGGGGCTTTCAGAACACAGGCATCTACAG-3' | | |
| rTRPV2 R535K | | |
| 5'-ACGGAGCAGGTCTTTAAGGATGACCTTCT<br>GGATCATGAC-3' | | |
| 5'-GTCATGATCCAGAAGGTCATCCTTAAAG<br>ACCTGCTCCGT-3' | | |
| rTRPV2 R539K | | |
| 5'-CCAGGTAGACCAGCAGGAACTTGAGCAGGTCTCGA<br>AGGATG-3' | | |
| 5'-CATCCTTCGAGACCTGCTCAAGTTCCTGCTGGTCT<br>ACCTGG-3' | | |
| rTRPV2-E495Q | | |
| 5'-GGTAGGTACCACTGAGTCTCCATGAAGCGCAGCAC-3' | | |
| **Chemicals, enzymes, and other reagents** | | |
| SD-Leu Media | Fisher BoiReagents | Cat# MP114811075 |
| Glycerol | Fisher BioReagents | Cat# BP229-4 |
| Protease inhibitor cocktail | Sigma-Aldrich | Cat# P8215 |
| CnBr-activated sepharose beads | Cytiva | Cat# 17043001 |
| 1D4 peptide | Genscript | N/A |
| Lauryl Maltose Neopentyl Glycol (LMNG) | Anatrace | Cat# NG310 |
| Decyl Maltose Neopentyl Glycol (DMNG) | Anatrace | Cat# NG322 |
| TCEP | Pierce | Cat# PG82090 |
| Soy polar lipids | Avanti | Cat# 541602C |
| MSP2N2 | Grinkova et al, 2010, Addgene | Cat# 29520 |
| Bio-Beads SM-2 Absorbent media | BioRad | Cat# 1528920 |
| diC8-PI(4,5)P2 | Echelon Biosciences | Cat# P-4508 |

| Reagent/resource | Reference or source | Identifier or catalog number |
|---|---|---|
| Ruthenium red | Sigma-Aldrich | Cat# 2751 |
| Nanofectin | PAA | Cat# |
| jetPEI | Polyplus-transfection | Cat# 13-101-10N |
| Effectene Transfection Reagent | Qiagen | Cat# 301425 |
| Poly-L-lysine | Sigma-Aldrich | Cat# P4707 |
| Dulbecco's modified Eagle medium | Gibco | Cat# 11995-065 |
| fetal bovine serum | Biochrom | Cat# F2442 |
| fetal bovine serum | Gibco | Cat# 12483-020 |
| formic acid | Sigma-Aldrich | Cat# 64-18-6 |
| lactic acid | Sigma-Aldrich | Cat# L1750 |
| propionic acid | Sigma-Aldrich | Cat# P1386 |
| benzoic acid | Sigma-Aldrich | Cat# 242381 |
| β-hydroxybutyric acid | Sigma-Aldrich | Cat# H6501 |
| acetic acid | Roth | Cat# 3738.4 |
| Latrunculin A | Cayman Chemical | Cat# 10010630 |
| Cytochalasin-D | Sigma-Aldrich | Cat# 8273 |
| Nocodazole | Sigma-Aldrich | Cat# 1404 |
| Cannabidiol | LGC GmbH | Cat#13956-29-1 |
| 2-APB | Tocris | Cat# 1224 |
| Ionomycin calcium salt | Tocris | Cat# 1704/1 |
| Pluronic F-127 | Biotium Inc | Cat# 59000 |
| BCECF-AM | Biotium Inc | Cat# 51010 |
| FURA2-AM | Biotium Inc | Cat# 50029 |
| **Software** | | |
| Patchmaster v2x90.5 | HEKA Electronik | https://www.heka.com/ |
| Relion-4.0 | Kimanius et al, 2021 | https://github.com/3dem/relion |
| cryoSPARC v3.3.x and v4.0.x | Punjani and Fleet, 2021; Punjani et al, 2020; Punjani et al, 2017 | https://cryosparc.com |
| COOT | Emsley et al, 2010 | https://www2.mrc-lmb.cam.ac.uk/personal/pemsley/coot |
| PHENIX | Afonine et al, 2018 | https://phenix-online.org/documentation/index.html |
| HOLE | Smart et al, 1996 | http://www.holeprogram.org/ |
| Chimera | Pettersen et al, 2004 | https://www.cgl.ucsf.edu/chimera/ |
| ChimeraX | Goddard et al, 2018 | https://www.cgl.ucsf.edu/chimerax/ |
| PyMol 2.3 | Schrodinger, Inc. | https://pymol.org/2/ |
| Origin 8.5.1 | Origin Lab | https://www.originlab.com/ |
| Excel | Microsoft | https://www.microsoft.com/en-us/microsoft-365/excel |
| VisiView 2.1.1 | Visitron Systems GmbH | https://www.visitron.de/ |
| **Other** | | |
| Alkali-Cation Yeast Transformation Kit | MP Biomedicals | Cat# 112200200 |
| QuikChange II XL Site-Directed Mutagenesis Kit | Agilent | Cat# 200522 |
| Quantifoil 1.2/1.3 grid | Quantifoil Micro Tools | |
| Superdex 200 column | Cytiva | Cat# 28990944 |

before use. Latrunculin A (Cayman Chemical, Ann Arbor, MI, USA), Cytochalasin-D and Nocodazole (Sigma-Aldrich, Taufkirchen, Germany) were stored as stock solutions at −20 °C. Cannabidiol (CBD) was obtained from LGC GmbH (Luckenwalde, Germany). 2-Aminoethoxydiphenylborane (2-APB) and Ionomycin calcium salt were purchased from Tocris (Bio-Techne GmbH, Wiesbaden, Germany). Ruthenium Red was obtained from Sigma-Aldrich (Taufkirchen, Germany).

## Cell culture

Human embryonic kidney cells (HEK293T) as well as murine/rat neuroblastoma derived-cell line (ND7/23) were used for patch clamp measurements or ratiometric Ca2 + -imaging. Cell cultures were treated with Dulbecco's modified eagle medium nutrient mixture F12 (DMEM/F12 Gibco/Invitrogen, Darmstadt, Germany) adding 10% fetal bovine serum (Biochrom, Berlin, Germany) and

1% penicillin/streptomycin (Lonza, Cologne, Germany). Cells were incubated under standard conditions at 5% carbon dioxide and 37 °C. Cells were cultured in 25 and 75 ml flasks (Sarstedt AG, Nümbrecht, Germany) and split every 3–4 days using Dulbecco's phosphate-buffered saline (PBS, Lonza, Cologne, Germany) and 0.05% trypsin (Biochrom GmbH, Berlin, Germany).

HEK293T cells or ND7/23 cells were transfected with TRPV2-plasmids using jetPEI (Polyplus-transfection® SA, Illkirch, France). cDNA of rTRPV2 was a kind gift from Dr. Michael Caterina (Johns Hopkins University School of Medicine, Baltimore). cDNA of mTRPV2 was a kind gift from Itaru Kojima (Gunma University, Maebashi, Japan). Cells were cultured in Dulbecco's modified Eagle medium nutrient mixture F12 (DMEM/F12 Gibco/Invitrogen, Darmstadt, Germany) supplemented with 10% fetal bovine serum (Biochrom, Berlin, Germany) under 5% CO2 at 37 °C. All mutants were generated by site-directed mutagenesis with the Quikchange lightning site-directed mutagenesis kit (Aglient, Waldbronn, Germany) according to the instructions of the manufacturer. All mutants were sequenced subsequently to exclude further channel mutation and to prove intended amino acid exchange. Cells were detached using phosphate-buffered saline (PBS, Lonza, Cologne, Germany) after ~24 h after transfection and seeded for experiments.

## Ratiometric pH measurements

Cells were seeded on coverslips stained for 45 min with 4-µM BCECF-AM. After washout, cells were mounted on an inverse microscope (Axio Observer D1; Zeiss, Jena, Germany). BCECF was excited at 500 and 436 nm using a light source (HXP 120, LEJ Lightning & Electronics, Jena, Germany), LEP filter wheel (Ludl Electronic Products Ltd., Hawthorne, NY, USA) and appropriate filter sets (Chroma Technology GmbH, Olching, Germany). Images were acquired at 1 Hz with a CCD camera (CoolSNAP EZ; Photometrics, Puchheim, Germany). Data were recorded with VisiView 2.1.1 software (Visitron Systems GmbH, Puchheim, Germany). Standard imaging solution (pH 7.4) contained (in mM) as follows: NaCl 145, KCl 5, CaCl$_2$ 1.25, MgCl$_2$ 1, glucose 10 and HEPES 10. Before the calculation of ratios, background fluorescence was subtracted. Results are presented as mean (± SEM) of the ratio F500/436 nm. The experimenter was blinded to the background and purpose of the experiment.

## Patch clamp electrophysiology

Borosilicate glass pipettes (GB 150 TF-8P, Science-products, Hofheim, Germany) fabricated with a Narishige PP-830 puller to a resistance between 2 and 5 Megaohm were used for whole-cell recordings. Inside-Out and on-Cell recordings were conducted with pipettes with a resistance of ~2 Megaohm. If not otherwise stated, cells were held at a constant membrane potential of −60 mV. Test solutions were applied to the dish next to the cell (<100 µM) by an additional pipette using a gravity-driven perfusion system (Dittert et al, 2006). This system also allows for heat-induced ion currents evoked by ~10 s long heat ramps applied by heating of the applied solution through a copper wire surrounding the tip of the perfusion pipette. HEKA Patchmaster software and EPC 10 USB HEKA amplifier (HEKA Elektronik, Lamprecht, Germany) were used for patch clamp recordings. Data were low-passed at 1 kHz and sampled at 10 kHz. HEKA FitMaster and Origin 8.5.1 (Origin Lab, Northampton, MA,

USA) were used for data analysis (HEKA Elektronik, Lamprecht, Germany). The intracellular solution contained (in mM): KCl 140, MgCl$_2$ 2, EGTA 5, HEPES 10 with pH adjusted to 7.4 by KOH. Nominal calcium-free extracellular solution contained (in mM): NaCl 140, KCl 5, MgCl$_2$ 2, EGTA 5 and HEPES 10; pH 7.4 adjusted by sodium hydroxide (NaOH). Calcium-containing solution contained (in mM): NaCl 140, KCl 5, MgCl$_2$ 2, CaCl$_2$ 2 and HEPES 10; pH 7.4 adjusted by sodium hydroxide (NaOH). For acidic pH solutions, we used 10 mM 2-(N-morpholino)ethanesulfonic acid ( = MES) as buffering solution instead of HEPES (10 mM). Excised patch recordings were performed using the inside-out patch clamp configuration with internal solution in the bath and external solution in the pipette. Single-channel recordings were performed and analyzed as described previously (Fricke et al, 2019). For analysis of ion permeability of rTRPV2, we adopted the approach described in previous reports (Caterina et al, 1997, 1999). In brief, the pipette solution contained (mM) 140 NaCl, 10 HEPES, 5 EGTA, pH 7.4. Permeability ratios for calcium and magnesium to Na (PX/PNa) were calculated as follows: PX/PNa = [Na + ]i exp(ΔVrevF/RT)(1 + exp(ΔVrevF/RT))/4[Y2 + ]O. Vrev is the reversal potential, F is Faraday's constant, R is the universal gas constant and T is absolute temperature. Assumed ion activity coefficients was 0.75 monovalent ions and 0.52 for divalent ions.

## Ratiometric Ca²⁺-imaging

Cells were stained with Fura-2 AM (3 µM) and 0.01% pluronic F-127 (both from Biotium Inc., Fremont CA, USA) for about 1 h. Coverslips were mounted on an inverse microscope. The extracellular solution contained (in mM): NaCl 145, KCl 5, CaCl$_2$ 1.25, MgCl$_2$ 1, glucose 10, HEPES 10 or MES 10) using a gravity-drived superfusion system. Fura-2 was excited using a microscope light source and an LEP filter wheel (Ludl Electronic Producs Ltd, Hathorne, USA) to switch between 340 and 380 nm. Images were exposed for 20 and 10 ms, respectively, and acquired at a rate of 1 Hz with a CCD camera (CoolSNAP EZ, Photometrics). Data were recorded using VisiView 2.1.1 software (all from Visitron Systems GmbH, Puchheim, Germany). Background fluorescence was subtracted before the calculation of ratios. Averaged results are reported as means (± SEM).

## Protein expression and purification

Full-length rat TRPV2 was expressed and purified as previously described (Fluck et al, 2021). Briefly, rat TRPV2 with a C-terminal 1D4 tag cloned into a Yep vector containing the PMA1 promotor was expressed in the BJ5457 strain of *Saccharomyces cerevisiae* (ATCC). These cells were resuspended in Homogenization Buffer (25 mM Tris, 300 mM Sucrose, 5 mM EDTA, pH 8.0) and lysed with a M110Y Microfluidizer (Microfluidics). The TRPV2-enriched membranes were separated by ultracentrifugation at 100,000 × g and then solubilized in Solubilization Buffer (20 mM Hepes, 150 mM NaCl, 5% glycerol, 0.087% LMNG, 2 mM TCEP, 1 mM PMSF, pH 8.0) for 1 h. The soluble material was separated from the insoluble material by ultracentrifugation at 100,000×g. The soluble material, enriched with TRPV2, was incubated with 1D4 antibody-coupled CnBr-activated Sepharose beads (Cytiva) for 2 h. The immobilized TRPV2 was washed with Wash Buffer (20 mM Hepes, 150 mM NaCl, 2 mM TCEP, pH 8.0) supplemented with 0.006% w/v DMNG and then eluted with Wash Buffer supplemented with 0.006% w/v DMNG and

3 mg/mL 1D4 peptide. The eluted TRPV2 was concentrated and reconstituted into nanodiscs in a 1:1:250 ratio of TRPV2 tetramer:MSP2N2:soy polar lipids (Avanti). The soy polar lipids were prepared by evaporating under a nitrogen flow, followed by resuspension in Wash Buffer containing DMNG in a 1:0.5 ratio of soy polar lipids:DMNG. The nanodisc reaction was incubated on ice for 30 min before the addition of Bio-Beads. After a 1 h incubation with Bio-Beads, the nanodisc reaction was transferred to a new tube with fresh Bio-Beads and incubated overnight. TRPV2 incorporated into nanodiscs was purified by size-exclusion chromatography using a Superose 6 column (GE) equilibrated with Wash Buffer. TRPV2 was concentrated to 2 mg/mL for use in cryo-EM grid preparation.

## Cryo-EM sample preparation and data collection

All conditions were prepared on freshly glow discharge 200 mesh copper Quantifoil 1.2/1.3 grids. 3 µL of the sample was applied at 4 °C and 100% humidity and blotted for 6 s in a Vitrobot Mark IV (Thermo Fisher) before vitrification in liquid ethane. All grids were prepared on the same day from the same preparation of TRPV2. For apo TRPV2, 1 mM fluorinated fos-choline 8 (Anatrace) was added immediately before plunge freezing. For the weak acid condition, 300 mM sodium acetate buffer at pH 4 was used as a 10× stock, which brought the sample pH to 5 and yielded a final concentration of 30 mM sodium acetate. The TRPV2 sample was exposed to the low pH condition for <5 min before plunge freezing. The condition with weak acid was also supplemented with 3 mM fluorinated fos-choline 8 (Anatrace) immediately before plunge freezing.

The weak acid dataset was collected in super-resolution mode on a 300 kV Thermo Fisher Krios equipped with a Gatan K3 direct detector camera with a super-resolution pixel size of 0.535 Å/pix. The apo dataset was collected on a 300 kV Thermo Fisher Krios equipped with a cold field-emission gun, a Selectris X energy filter, and a Falcon4i detector (Thermo Fisher) at a nominal magnification of ×165,000 with a resulting pixel size of 0.73 Å/pixel. Dose-rate measured on the detector over vacuum at spot size 4 at 650 nm illumination was 7.43 electrons/pixel/second and each exposure was 3.54 s long to accumulate a total dose of 50 electrons/Å$^2$ per exposure. Images were collected at defocus values ranging from −0.6 µm to −1.4 µm in steps of 0.2 µm.

## Cryo-EM data processing

For the Apo dataset, all processing was done in Relion-4.0-beta (Kimanius et al, 2021). Overall, 14,010 movies were aligned using the Relion implementation of MotionCor2 (Zheng et al, 2017). The defocus values of the resulting micrographs were estimated using CTFFIND-4.1 (Rohou and Grigorieff, 2015). In total, 652,062 particles were picked by Topaz, using a picking model trained from the dataset. The particles were subjected to two rounds of 2D classification to remove false positives and bad particles, using particles binned by 4 for the first round and binned by 2 for the second round, resulting in 379,687 good particles. The nanodisc density for these particles is pronounced and dominated initial classification, therefore, the particles were first refined without applied symmetry using our previously published apo rat TRPV2 (EMD-20677) as a reference model and then density for the nanodiscs was subtracted. The resulting subtracted particles were then subjected to 3D classification with angular assignment, resulting in 233,663 particles with good, unbroken channels. The subtracted particles were then reverted to their original form and refined again without symmetry, followed by another

round of 3D classification with local angular assignment and relaxed C4 symmetry. This resulted in classes reflecting the three final states: 39,326 particles in a class with an open selectivity filter for the Apo2 state, 36,155 particles in a class with a closed selectivity filter and C-terminal helix for the Apo1 state, and 37,063 particles in a class with a closed selectivity filter and a C-terminal loop for the Apo3 state. The 39,326 particles in the Apo2 class were re-extracted at their final pixel size, 0.73 Å/pix, and refined with C4 symmetry, followed by CTF refinement (Zivanov et al, 2020), Bayesian polishing (Zivanov et al, 2019), and another round of refinement with C4 symmetry. These particles were then subjected to another round of classification without angular assignment, yielding a final class made up of 17,515 particles for TRPV2Apo2. The 36,155 particles in the Apo1 class were sorted again with another round of 3D classification with local angular assignment and relaxed C4 symmetry, resulting in two classes that could be combined for a final set of 23,986 particles for TRPV2Apo1. These 23,986 particles were re-extracted at their final pixel size and then refined with C4 symmetry, followed by CTF refinement, Bayesian polishing, and another round of refinement with C4 symmetry. The 37,063 particles in the Apo3 state were sorted again with another round of 3D classification with local angular assignment and relaxed C4 symmetry, resulting in two classes that could be combined for a set of 25,160 particles. These particles were then re-extracted at their final pixel size and then refined with C4 symmetry, followed by CTF refinement, Bayesian polishing, and another round of refinement with C4 symmetry. They were then subjected to a round of 3D classification without angular assignment, yielding a final set of 9,483 particles for TRPV2Apo3. All of the refined maps were sharpened with the Phenix tool resolve_cryo_em model-based map sharpening (Liebschner et al, 2019).

For the weak acid dataset, initial processing was done in cryoSPARC v3 (Punjani et al, 2017). Overall, 11,071 movies were aligned using Patch Motion Correction, and the defocus values of the resulting micrographs were estimated using Patch CTF estimation. The dataset was then curated to remove suboptimal micrographs, resulting in a set of 10,200 good micrographs. 658,222 particles were picked by Topaz (Bepler et al, 2020), using a picking model trained from the dataset. The particles were subjected to two rounds of 2D classification to remove false positives and bad particles, using particles binned by 4 for the first round and binned by 2 for the second round, resulting in 460,659 good particles. To initially reconstruct the three-dimensional channel and sort out more suboptimal particles, the 460,659 good particles were subjected to three consecutive rounds of hetero refinement, in each case using one copy of a good reference (our previously published apo TRPV2, EMD-20677) and three copies of an unrelated reference, resulting in 412,537 particles that consistently sorted into the good class. These particles then underwent non-uniform refinement (Punjani et al, 2020), including refinement of CTF parameters (Zivanov et al, 2020), without applied symmetry. The nanodisc density for these particles is pronounced and dominated initial attempts at classification, therefore the nanodisc density for the aligned particles was subtracted before proceeding to another round of hetero refinement, this time using three copies of the refined TRPV2 as references. After sorting out broken TRPV2 particles, 307,276 particles remained. The dataset had a single overall state, with an open selectivity filter and the C-terminus in a helical conformation. To minimize heterogeneity, the 307,276 particles were refined without symmetry with non-uniform refinement before undergoing 3D variability analysis using a mask focused on the transmembrane domain (Punjani and Fleet,

2021). This revealed a single primary mode, transitioning between particles with a broken pore domain and intact particles. We used 3D variability display intermediate mode analysis to section the particles into 10 clusters along this mode and chose particles from the intact end (clusters 1 and 2) to combine, resulting in the final set of 62,824 particles for TRPV2$_{HOAc}$. These particles were reverted to their original unsubtracted form and refined with C4 symmetry using non-uniform refinement.

## Model building

The initial models used to fit into each map were: PDB 6U84 for TRPV2$_{Apo1}$, PDB 6U86 for TRPV2$_{Apo2}$, PDB 6U88 for TRPV2$_{Apo3}$, and PDB 6U86 for TRPV2$_{HOAc}$. Models were manually adjusted to the maps in Coot (Emsley et al, 2010) and then refined using phenix.real_space_refine (Afonine et al, 2018). Pore profiles were made with Hole. Images were rendered with Pymol (The PyMOL Molecular Graphics System, Version 2.0 Schrödinger, LLC), Chimera (Pettersen et al, 2004) and ChimeraX (Goddard et al, 2018). Sequence alignment was done with Clustal Omega (Sievers et al, 2011), and the figure was produced with ALINE (Bond and Schuttelkopf, 2009).

## Molecular modeling and simulations

The pKa of the identified pH-sensitive residues within the resolved TRPV2 structures were predicted using PropKa3 (Olsson et al, 2011). Following this, the conformations of the unresolved loop regions were obtained from AlphaFold (Varadi et al, 2022) and modeled onto the resolved TRPV2Apo1 structure. The constructed structures were then embedded in a POPC bilayer, solvated in TIP3P water and 0.15 M NaCl/KCl using Charmm-GUI (Wu et al, 2014). For each structure, two simulation systems were built either with or without protonation of the His521, Glu561, Glu599, and His651 residues. Protein, lipids, and ions were described using the Charmm36M force field (Huang et al, 2017). Long-range electrostatic interactions were calculated using the particle mesh Ewald method and hydrogen bond lengths were constrained using LINCS (Hess, 2008). Pressure (1 bar) and temperature (300 K) were maintained using the semi-isotropic Parrinello-Rahman (Parrinello and Rahman, 1981) barostat and the v-rescale thermostat (Bussi, Donadio, Parrinello, 2007) respectively. After energy minimization and equilibration, four replicas of each system were simulated for 300 ns each using Gromacs 2021.3 (Abraham et al, 2015) and a timestep of 2 fs. The simulation trajectories were analyzed using the MDAnalysis library (Michaud-Agrawal, Denning et al, 2011).

## Weak acid-induced cell death

Cell death induced by pH 5.0 titrated with HCl or by 10 and 30 mM HOAc at pH 5.0 was examined on untransfected HEK293 cells as well as on HEK293 stably expressing rTRPV2. Acids were applied for 2 h at 37 °C in the buffer used for patch clamp recordings. Cells were stained with propidium iodide (PI, 10 µg/ml, Thermo Fischer) for 20 min at 37 °C, fixed with 4% paraformaldehyde for 20 min, and counterstained with 4′,6-diamidino-2-phenylindole (DAPI, 1 µg/ml in PBS supplemented with 1% bovine serum albumin) for 20 min. For quantification, 5 10× microscopic fields were randomly chosen on an inverted fluorescence microscope (Olympus IX81). The percentage of PI positive cells over the total number of cells, identified with DAPI, was determined using Image J (NIH).

## Statistical analysis

Statistical analysis was performed using Origin 8.5.1 (Origin Lab, Northampton, MA, USA). More than two groups were compared using ANOVA, followed by the Bonferroni correction. If not otherwise noted, significance was assumed for $P < 0.05$.

## Data availability

The datasets produced in this study are available in the following databases: The atomic coordinates and cryo-EM density maps generated in this study have been deposited in the Protein Data Bank and Electron Microscopy Data Bank under the accession codes PDB 8EKP and EMD-28209 for TRPV2Apo1, PDB 8EKQ and EMD-28210 for TRPV2Apo2, PDB 8EKR and EMD-28211 for TRPV2Apo3, and PDB 8EKS and EMD-28212 for TRPV2pH5. The MD simulation datasets generated for this study can be accessed under: https://doi.org/10.5281/zenodo.10018876.

The source data of this paper are collected in the following database record: biostudies:S-SCDT-10_1038-S44318-024-00106-4.

## Peer review information

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

## Acknowledgements

The authors thank Mrs. Heike Bürger, Mrs. Kerstin Gutt, and Ms. Marina Golombek (all Hannover Medical School) for excellent technical assistance. The authors acknowledge the use of instruments at the Electron Microscopy Resource Lab and at the Beckman Center for Cryo-Electron Microscopy at the University of Pennsylvania Perelman School of Medicine. We also thank Stefan Steimle for assistance with Krios microscope operation at the University of Pennsylvania. We thank Sabine Baxter for assistance with hybridoma and cell culture at the University of Pennsylvania Perelman School of Medicine Cell Center Services Facility. AS was supported by Marie–Sklodowska–Curie grant 898762. EL and RJH were supported by the Swedish Research Council (2019-02433, 2021-05806), the Swedish e-Science Research Centre, and the BioExcel Center of Excellence (EU-823830). Computational resources were provided by the Swedish National Infrastructure for Computing. This work was supported by grants from the National Institute of Health (R01GM103899 and R01GM129357 to VYM-B). AL was supported by the Friedrich und Alida Gehrke-Stiftung.

## Author contributions

**Ferdinand M Haug**: Formal analysis; Investigation; Methodology; Writing—original draft. **Ruth A Pumroy**: Data curation; Formal analysis; Investigation; Visualization; Writing—original draft. **Akshay Sridhar**: Formal analysis; Investigation; Visualization; Writing—original draft. **Sebastian Pantke**: Formal analysis; Visualization. **Florian Dimek**: Formal analysis; Investigation; Visualization. **Tabea C Fricke**: Formal analysis; Investigation; Visualization. **Axel Hage**: Formal analysis; Investigation; Visualization. **Christine Herzog**: Resources; Investigation; Methodology. **Frank G Echtermeyer**: Investigation; Visualization; Methodology. **Jeanne de la Roche**: Formal analysis; Investigation; Visualization; Methodology; Writing—original draft. **Adrian Koh**: Resources; Software. **Abhay Kotecha**: Resources; Software. **Rebecca J Howard**: Formal analysis; Validation; Investigation; Visualization. **Erik Lindahl**: Resources; Supervision. **Vera Moiseenkova-Bell**: Formal analysis; Validation; Visualization; Methodology; Writing—review and editing. **Andreas Leffler**: Conceptualization; Resources; Data curation; Formal analysis; Supervision; Funding acquisition; Validation; Investigation; Visualization; Methodology; Project administration; Writing—review and editing.

Source data underlying figure panels in this paper may have individual authorship assigned. Where available, figure panel/source data authorship is listed in the following database record: biostudies:S-SCDT-10_1038-S44318-024-00106-4.

## Funding

## Disclosure and competing interests statement

The authors declare no competing interests.

# Expanded View Figures

**Figure EV1.  Weak acids inhibit endogenous PAC-mediated currents in HEK293T cells.**

(**A**) BCECF-based ratiometric imaging on non-transfected HEK293T cells treated with pH 5.0 titrated with HCl, or 10 or 30 mM HOAc at pH 5.0. Application of HOAc, and to a lesser degree pH 5.0 induced a reversible decrease in fluorescence ratio, correlating with an intracellular acidosis. (**B**) Representative current trace from a non-transfected HEK293T cell demonstrating that 30 mM HOAc at pH 5.0 fails to evoke slowly activating currents without TRPV2 in the cell. Note the very rapidly inactivating currents emerging at the beginning of the recordings, demonstrating proton-evoked activation of endogenous acid-sensitive ion channels (ASIC1a) in some HEK293T cells. (**C**) Membrane currents in non-transfected HEK293T cells challenged with pH 5.0 or the combination of pH 5.0 and 100 μM pregnenolone sulfate. (**D**) Box diagrams with dot plots displaying normalized currents at +100 mV from cells described under C ($n = 5$). (**E**) Membrane currents in non-transfected HEK293T cells challenged with pH 5.0 or the combination of pH 5.0 ($n = 9$) and 10 ($n = 5$) or 30 mM ($n = 9$) HOAc. (**F**) Box diagrams with dot plots displaying normalized currents at +100 mV from cells described under (**E**). (**C, E, G**) Currents were monitored during a 500 ms long voltage ramp ranging from −100 mV to +100 mV. (**G**) Voltage-dependent membrane currents in a non-transfected HEK293T cell evoked by 200 ms long pulses from −60 to 120 mV applied in steps of 20 mV. Currents were evoked in control solution and in 10 mM HOAc at p 5.0. (**H**) Current–voltage plots of experiments performed in (**G**) ($n = 4$). Peak current amplitudes were normalized to the amplitude evoked at 120 mV. (**I, J**) Current traces on rTRPV2-expressing cells with two consecutive applications of 200 μM 2-APB. For the 2nd application, 2-APB was combined with pH 5.0 with HCl (**I**) or with 10 mM HOAC at pH 5.0 (**J**). (**K**) Box diagrams with dot plots displaying normalized current amplitudes of currents in induced by 2-APB (set as 1), 2-APB + pH 5.0 ($n = 8$) or 2-APB + HOAc ($n = 8$). (**D, F, K**) The box denotes the 50th percentile (median) as well as the 25th and 75th percentile. The whiskers mark the 5th and 95 percentiles. Data points beyond the whiskers are outliers. Source data are available online for this figure.

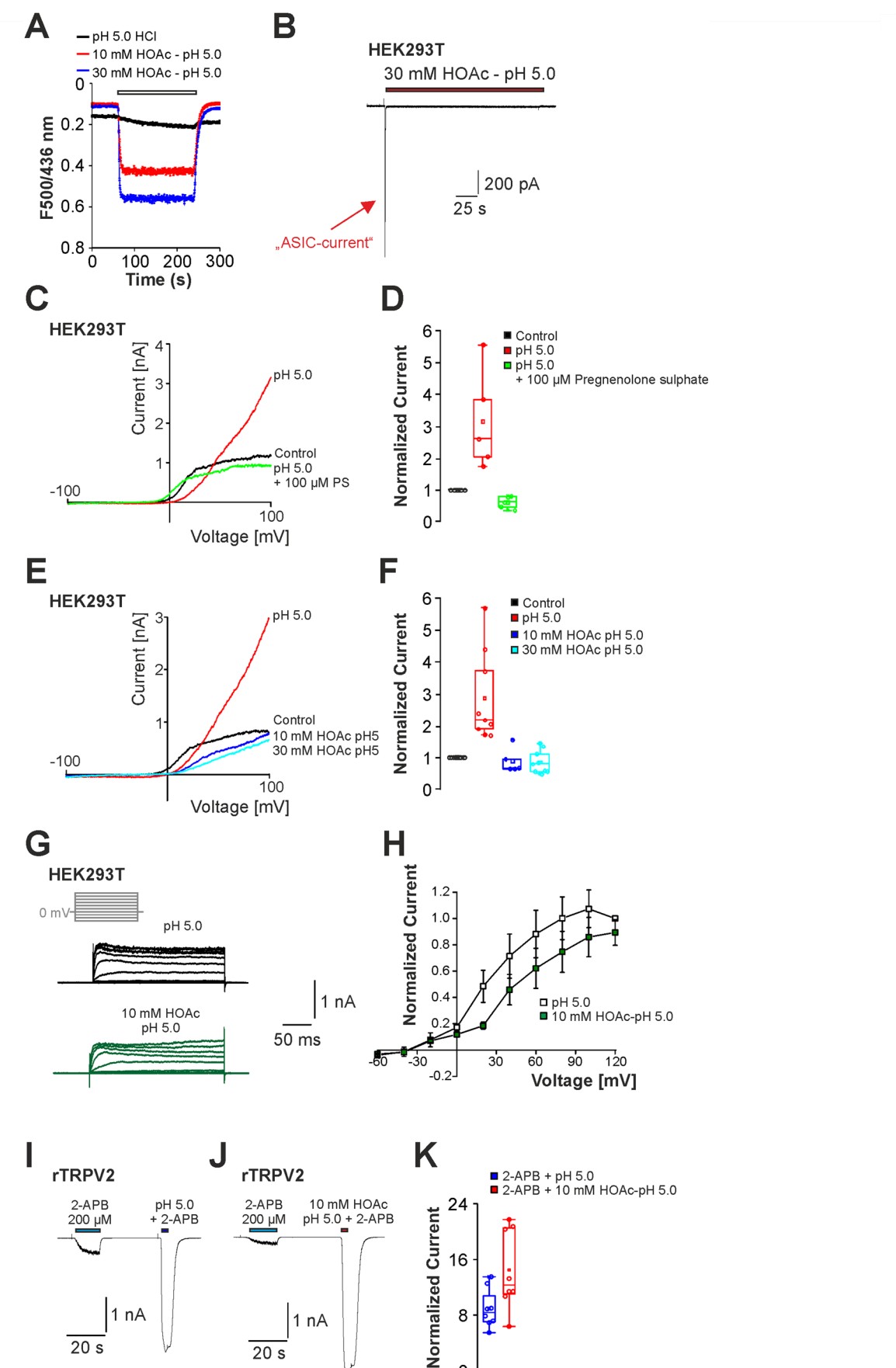

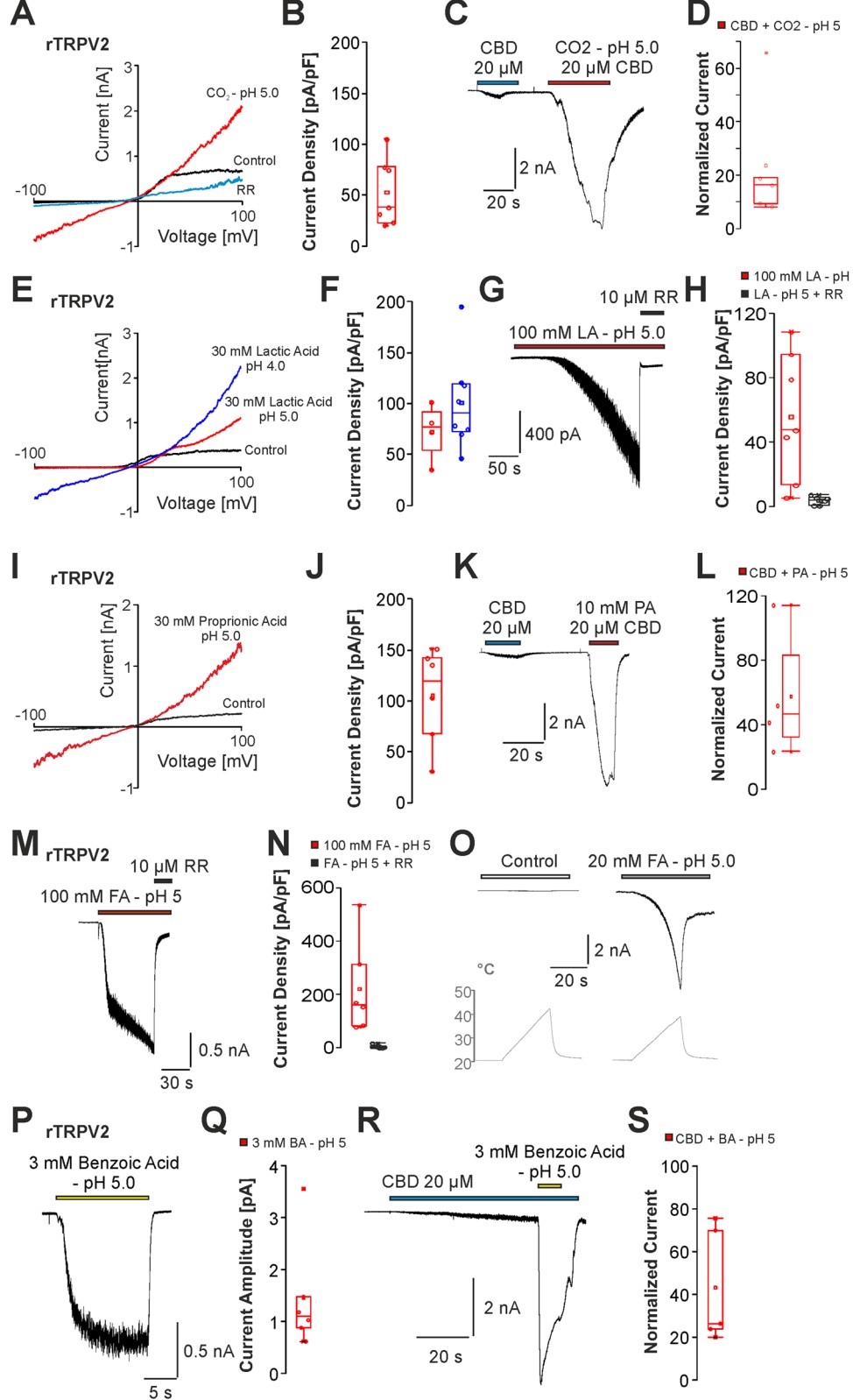

**Figure EV2. Several weak acids activate TRPV2.**

(A) Membrane currents observed in cells expressing rTRPV2 challenged with $CO_2$ at pH 5.0. (B) Box diagrams with dot plots displaying the mean current densities at +100 mV induced by $CO_2$ ($n = 7$). (C) Current trace from a rTRPV2-expressing cell generating inward currents induced by CBD or CBD in combination with $CO_2$. (D) Box diagrams with dot plots displaying the degree of potentiation induced by $CO_2$ ($n = 5$) The $CO_2$-solution was achieved by bubbling the extracellular solution with $CO_2$ using a commercially available soda streamer. (E) Membrane currents induced by 30 mM lactic acid at pH 5.0 ($n = 4$) and 4.0 ($n = 8$). (F) Box diagrams with dot plots displaying the mean current densities at +100 mV induced by lactate. (G) Activation of an inward current in a rTRPV2-expressing cell exposed to 100 mM lactate at pH 5.0. The inward current was blocked by co-application with 10 µM ruthenium red (RR). (H) Box diagrams with dot plots displaying the current densities of lactate-induced inward currents in rTRPV2-expressing cells ($n = 7$). (I) Membrane currents induced by 30 mM propionic acid at pH 5.0 on rTRPV2. (J) Box diagram with dot plots displaying the mean current densities at +100 mV induced by propionic acid ($n = 6$). (K) Current trace from a rTRPV2-expressing cell generating inward currents induced by CBD or CBD in combination with propionic acid at pH 5.0. (L) Box diagram with dot plots displaying the degree of potentiation induced by propionic acid. (M) Inward current induced by 100 mM formic acid (FA) at pH 5.0 on rTRPV2 ($n = 4$). The inward current was blocked by 10 µM ruthenium red (RR). (N) Box diagrams with dot plots displaying the current densities of formic acid-induced inward currents in rTRPV2-expressing cells ($n = 6$). (O) Representative traces displaying heat-evoked currents induced by 20 mM formic acid at pH 5.0 on a rTRPV2-expressing cell. (P) Inward current induced by 3 mM bezoic acid at pH 5.0 on rTRPV2. (Q) Box diagrams with dot plots displaying the current densities of benzoic acid-induced inward currents ($n = 6$). (R) Original trace from a rTRPV2-expressing cell generating inward currents induced by CBD or CBD in combination with 3 mM benzoic acid at pH 5.0. (S) Box diagram with dot plots displaying the degree of potentiation induced by benzoic acid ($n = 5$). In (B, D, F, H, J, L, N, Q, S), the box denotes the 50th percentile (median) as well as the 25th and 75th percentile. The whiskers mark the 5th and 95 percentiles. Data points beyond the whiskers are outliers. Source data are available online for this figure.

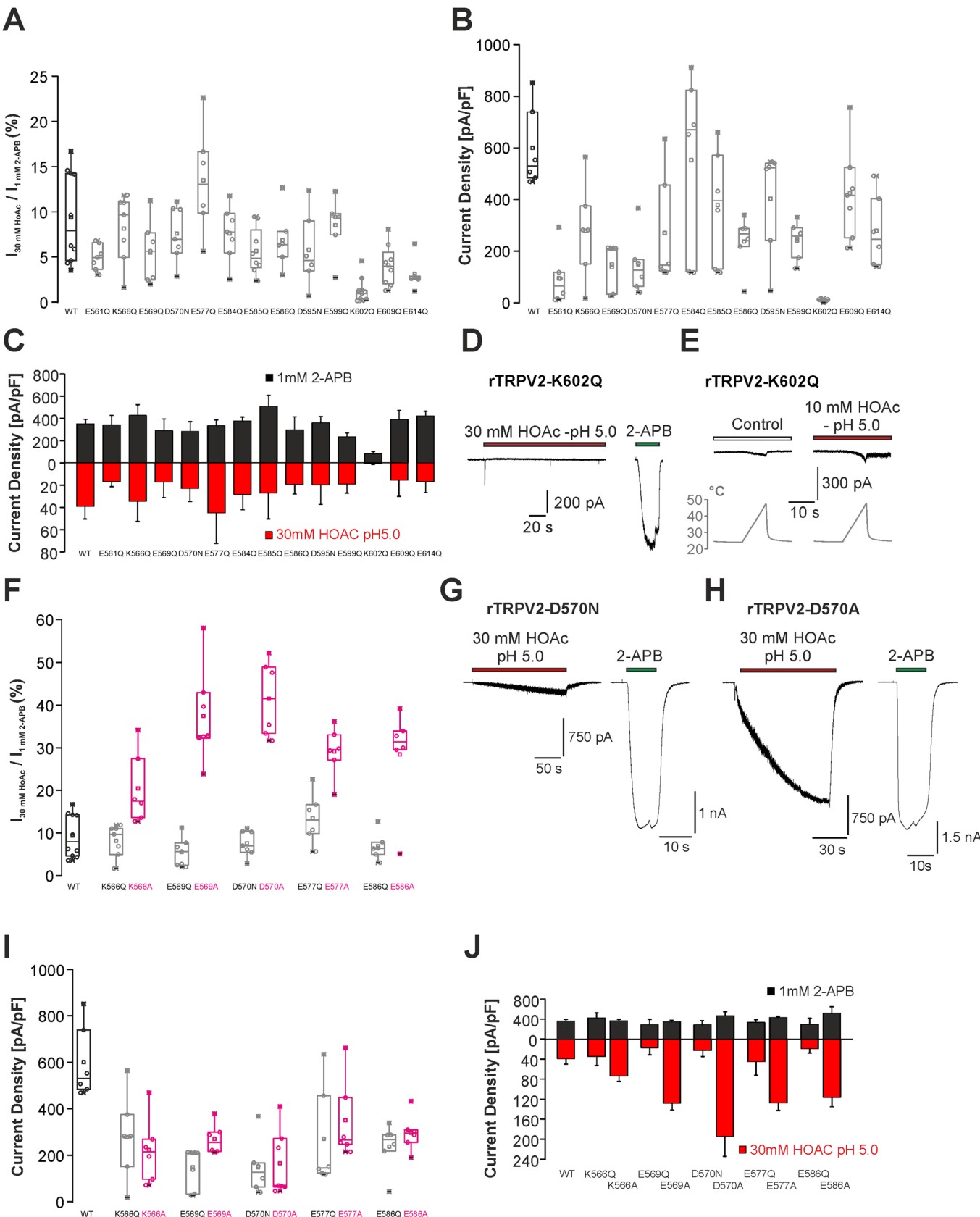

◄ **Figure EV3. The role of pore residues for weak acid-sensitivity of TRPV2.**

(A) Box diagrams with dot plots displaying the relative magnitudes of inward currents induced by 30 mM HOAc at pH 5.0 normalized with the responses evoked by 1 mM 2-APB in the corresponding cells ($n = 5$–10 for each mutant). (B) Box diagrams with dot plots displaying the current densities of heat-evoked currents provoked by 10 mM HOAc at pH 5.0 ($n = 6$–7 for each mutant). (C) Bar diagrams displaying mean ( ± SEM) current densities for currents evoked by 30 mM HOAc at pH 5.0 or 1 mM 2-APB ($n = 5$–8 for each mutant). (A–C) Data are demonstrated for rTRPV2-WT, -E561Q, -K566Q, -E569Q, -D570N, -E577Q, -E584Q, -E585Q, -E586Q, -D595N, -E599Q, -K602Q, -E609Q and −614Q. (D) Example of a current trace displaying that 30 mM HOAc at pH 5.0 fails to activate rTRPV2-K602Q. Note that the inward current evoked by 1 mM 2-APB is relatively small. (E) Heat-evoked currents in a cell expressing rTRPV2-K602Q. Note that 10 mM HOAc at pH 5.0 only provokes a minimal heat-evoked current. (F) Box diagrams with dot plots displaying the relative magnitudes of HOAc-evoked inward currents normalized with the responses evoked by 1 mM 2-APB ($n = 6$–7 for each mutant). Data are displayed for rTRPV2-WT, -K566Q, -K566A, -E569Q, -E569A, -D570N, -D570A -E577Q, -E577A, -E586Q and -E586A. (G, H) Representative current traces from HEK293T cells expressing rTRPV2-D570N (G) and rTRPV2-D570A (H). 30 mM HOAc at pH 5.0 evoked large currents in cells expressing rTRPV2-D570A. Functionality and expression was validated by application of 1 mM 2-APB. (I) Box diagrams with dot plots displaying the current densities of heat-evoked currents provoked by 10 mM HOAc at pH 5.0 in cells expressing rTRPV2-WT, -K566Q, -K566A, -E569Q, -E569A, -D570N, -D570A -E577Q, -E577A, -E586Q and -E586A ($n = 6$–7 for each mutant). (J) Bar diagrams displaying mean ( ± SEM) current densities for currents evoked by 30 mM HOAc at pH 5.0 or 1 mM 2-APB in cells expressing rTRPV2-WT, -K566Q, -K566A, -E569Q, -E569A, -D570N, -D570A -E577Q, -E577A, -E586Q and -E586A ($n = 6$–7 for each mutant). (A, B, F, I) The box denotes the 50th percentile (median) as well as the 25th and 75th percentile. The whiskers mark the 5th and 95 percentiles. Data points beyond the whiskers are outliers. Source data are available online for this figure.

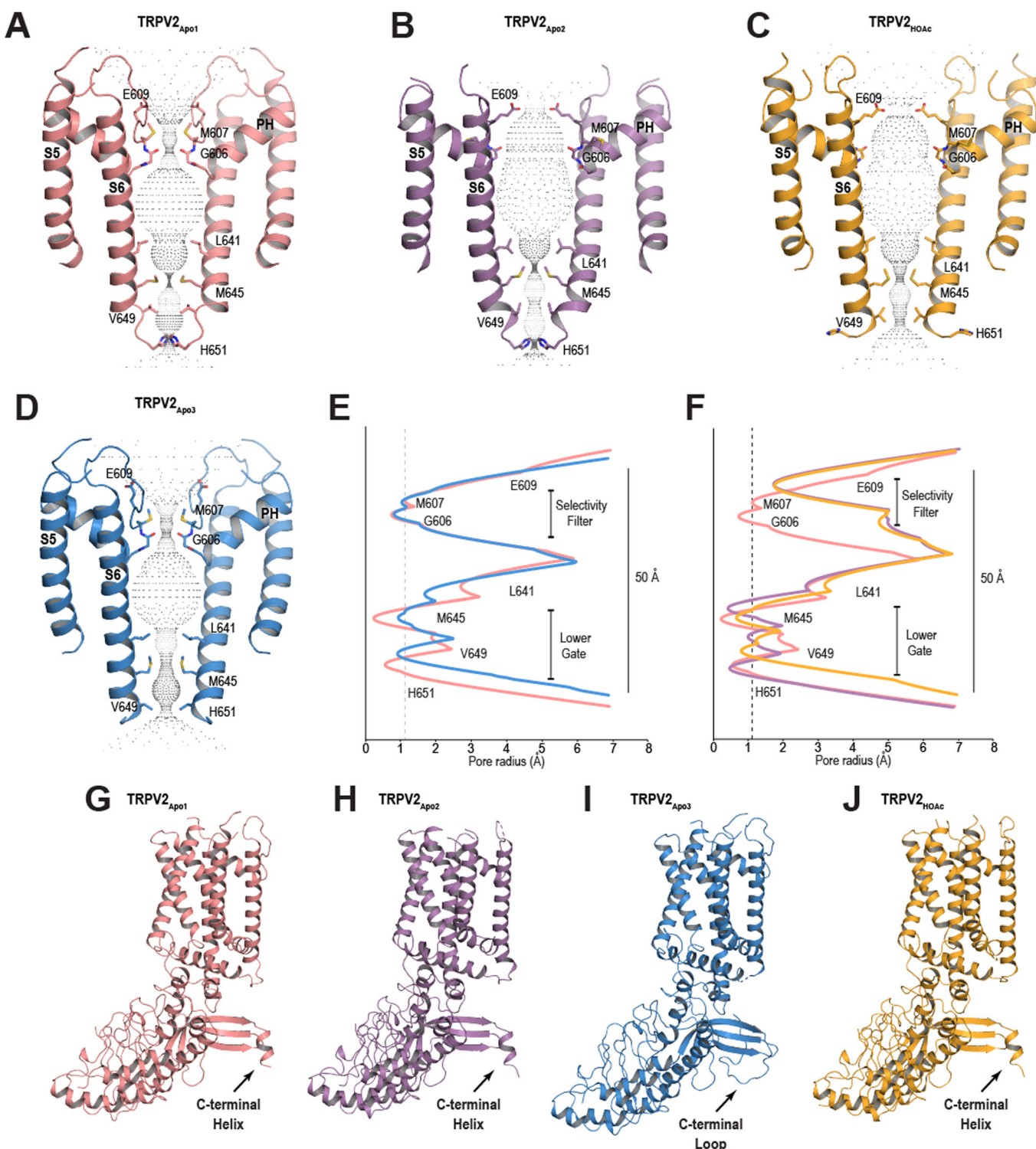

**Figure EV4. The effect of HOAc on the pore profile of rTRPV2.**

(A–D) Pore profiles of TRPV2Apo1 (A), TRPV2Apo2 (B), TRPV2HOAc (C), and TRPV2Apo3 (D). (E, F) A graphical representation comparing the pore diameter of TRPV2Apo3 (blue) to TRPV2Apo1 (salmon) (E) or TRPV2HOAc (orange) to TRPV2Apo2 (purple) and TRPV2Apo1 (salmon) (F). The dotted line marks the radius of a dehydrated calcium ion. (G–J) Side view of a single monomer of TRPV2Apo1 (G), TRPV2Apo2 (H), TRPV2Apo3 (I), and TRPV2HOAc (J). Arrows indicate C-terminus.

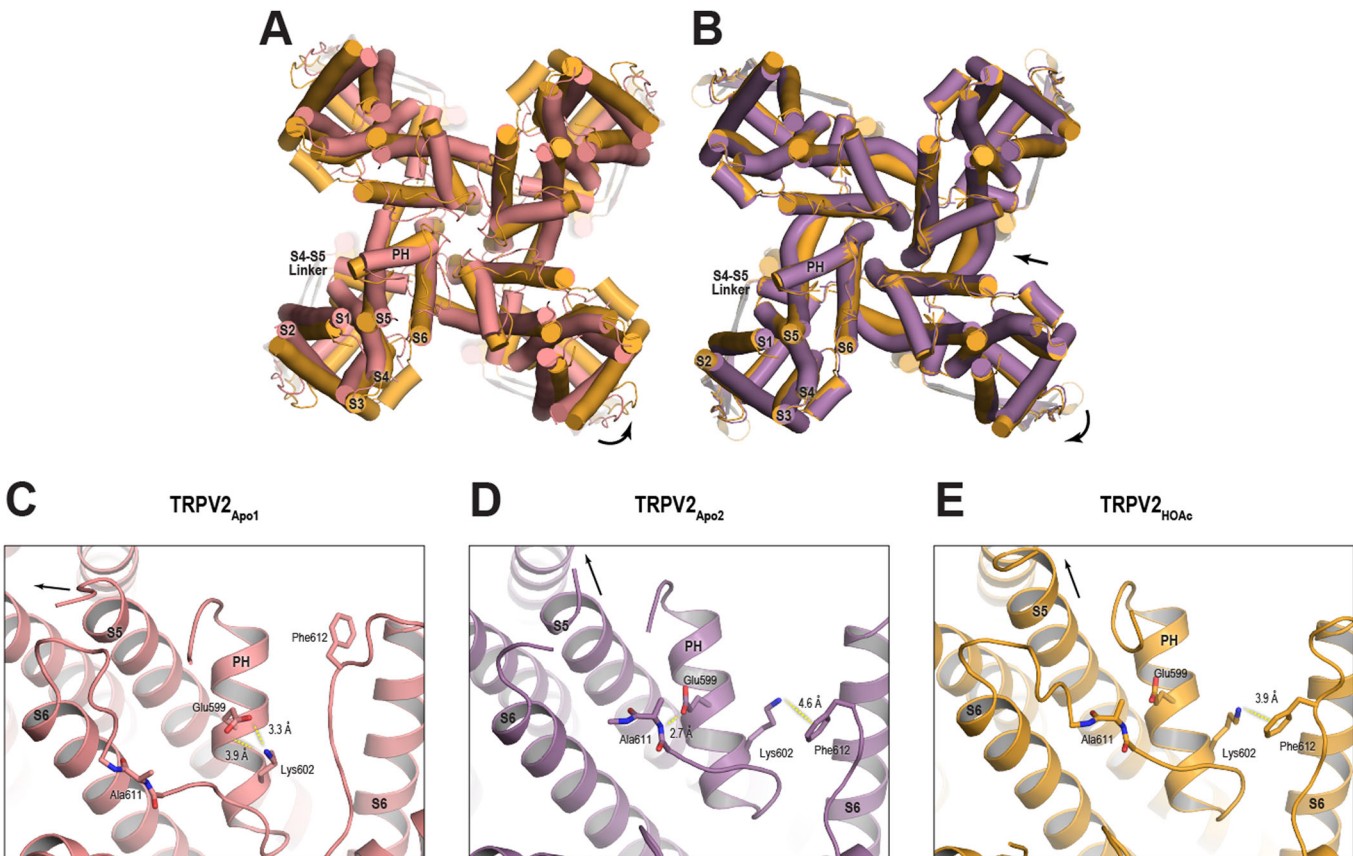

**Figure EV5.   HOAc-induced movements of rTRPV2.**

(A, B). Alignments of TRPV2$_{Apo1}$ (salmon) vs. TRPV2$_{HOAc}$ (orange) (A) or TRPV2$_{Apo2}$ (purple) vs. TRPV2$_{HOAc}$ (orange) (B) at the extracellular face of the channel. Arrows indicate directions of movement in the transition from the apo states to TRPV2$_{HOAc}$. (C–E) View the pore helix in TRPV2$_{Apo1}$ (C), TRPV2$_{Apo2}$ (D), and TRPV2$_{HOAc}$ (E). Bonds between residues are indicated by a yellow dashed line. Arrows indicate trajectory of the top of S5.

