## [Peer Review File · The EMBO Journal]

Functional and structural insights into activation of TRPV2 by weak acids

Ferdinand Haug, Ruth Pumroy, Aksahay Sridhar, Sebastian Pantke, Florian Dimek, Tabea Fricke, Axel Hage, Christine Herzog, Frank Echtermeyer, Jeanne de la Roche, Adrian Koh, Abhay Kotecha, Rebecca Howard, Erik Lindahl, Vera Moiseenkova-Bell, and Andreas Leffler

Corresponding authors: Andreas Leffler (leffler.andreas@mh-hannover.de) , Vera Moiseenkova-Bell (vmb@pennmedicine.upenn.edu)

Review Timeline:

Submission Date:	2nd Oct 23
Editorial Decision:	22nd Nov 23
Revision Received:	7th Feb 24
Editorial Decision:	8th Mar 24
Revision Received:	25th Mar 24
Accepted:	27th Mar 24

Editor: Daniel Klimmeck

Transaction Report:

Dear Dr Leffler,

Thank you again for submitting your manuscript EMBOJ-2023-115715 for consideration by the EMBO Journal. Please accept my apologies for getting back to you with protraction due to delayed referee input, as well as detailed discussion in the editorial team. As indicated, your manuscript has been seen by three referees with expertise in channel and structural biology, and we have received reports from all of them, which are shown below.

Given the referees' positive recommendations, I would like to invite you to submit a revised version of the manuscript, addressing the comments of all three experts. I should add that it is EMBO Journal policy to allow only a single round of revision, and acceptance of your manuscript will therefore depend on the completeness of your responses in this revised version.

I would appreciate if you could contact me during the next weeks for exchange e.g. a video call to discuss your perspective on the comments and potential plan for revisions.

When submitting your revised manuscript, please carefully review the instructions below.

Please feel free to approach me any time should you have additional questions related to this.

Thank you for the opportunity to consider your work for publication.

I look forward to your revision.

Kind regards,

Daniel Klimmeck

Daniel Klimmeck, PhD
Senior Editor
The EMBO Journal

Instruction for the preparation of your revised manuscript:

- 1) a .docx formatted version of the manuscript text (including legends for main figures, EV figures and tables). Please make sure that the changes are highlighted to be clearly visible.
- 2) individual production quality figure files as .eps, .tif, .jpg (one file per figure).
- 3) a .docx formatted letter INCLUDING the reviewers' reports and your detailed point-by-point response to their comments. As part of the EMBO Press transparent editorial process, the point-by-point response is part of the Review Process File (RPF), which will be published alongside your paper.
- 4) a complete author checklist, which you can download from our author guidelines ([https://wol-prod-cdn.literatumonline.com/pb-assets/embo-site/Author Checklist%20-%20EMBO%20J-1561436015657.xlsx](https://wol-prod-cdn.literatumonline.com/pb-assets/embo-site/Author%20Checklist%20-%20EMBO%20J-1561436015657.xlsx)). Please insert information in the checklist that is also reflected in the manuscript. The completed author checklist will also be part of the RPF.
- 5) Please note that all corresponding authors are required to supply an ORCID ID for their name upon submission of a revised manuscript.
- 6) It is mandatory to include a 'Data Availability' section after the Materials and Methods. Before submitting your revision, primary datasets produced in this study need to be deposited in an appropriate public database, and the accession numbers and database listed under 'Data Availability'. Please remember to provide a reviewer password if the datasets are not yet public (see

<https://www.embopress.org/page/journal/14602075/authorguide#datadeposition>).

7) Our journal encourages inclusion of *data citations in the reference list* to directly cite datasets that were re-used and obtained from public databases. Data citations in the article text are distinct from normal bibliographical citations and should directly link to the database records from which the data can be accessed. In the main text, data citations are formatted as follows: "Data ref: Smith et al, 2001" or "Data ref: NCBI Sequence Read Archive PRJNA342805, 2017". In the Reference list, data citations must be labelled with "[DATASET]". A data reference must provide the database name, accession number/identifiers and a resolvable link to the landing page from which the data can be accessed at the end of the reference. Further instructions are available at .

8) At EMBO Press we ask authors to provide source data for the main and EV figures. Our source data coordinator will contact you to discuss which figure panels we would need source data for and will also provide you with helpful tips on how to upload and organize the files.

Numerical data can be provided as individual .xls or .csv files (including a tab describing the data). For 'blots' or microscopy, uncropped images should be submitted (using a zip archive or a single pdf per main figure if multiple images need to be supplied for one panel). Additional information on source data and instruction on how to label the files are available at .

9) We replaced Supplementary Information with Expanded View (EV) Figures and Tables that are collapsible/expandable online (see examples in <https://www.embopress.org/doi/10.15252/emboj.201695874>). A maximum of 5 EV Figures can be typeset. EV Figures should be cited as 'Figure EV1, Figure EV2' etc. in the text and their respective legends should be included in the main text after the legends of regular figures.

11) For data quantification: please specify the name of the statistical test used to generate error bars and P values, the number (n) of independent experiments (specify technical or biological replicates) underlying each data point and the test used to calculate p-values in each figure legend. The figure legends should contain a basic description of n, P and the test applied. Graphs must include a description of the bars and the error bars (s.d., s.e.m.).

We realize that it is difficult to revise to a specific deadline. In the interest of protecting the conceptual advance provided by the work, we recommend a revision within 3 months (20th Feb 2024). Please discuss the revision progress ahead of this time with the editor if you require more time to complete the revisions. Use the link below to submit your revision:

Referee #1:

Haug et al. clearly showed that weak acids such as acetic acid and lactic acid activate and sensitize TRPV2 through a mechanism requiring permeation of weak acids through the cell membrane. And the authors clarified the structural basis of the weak acids-induced TRPV2 activation with a cryo-EM and MD simulation. The presented results look very solid and support the conclusion.

This reviewer wants to ask the authors to do some experiments to strengthen their logic.

1. The authors described a mechanism requiring permeation of weak acids through the cell membrane. It would be possible to estimate the cytosolic pH changes beneath the plasma membrane upon application of weak acids to the extracellular solution. The data should be compared with the proton (inside the cell) dose-dependency for the activation of TRPV2.

2. The authors showed the weak acid insensitivity of human TRPV2. Comparison of the TRPV2 structure with acetic acids by Cryo-EM between rat TRPV2 and human TRPV2 would support the structural basis of the weak acid-induced activation of TRPV2.

3. The reviewer wants to know the changes in temperature thresholds for heat-evoked activation of rat TRPV2 in the presence of HOAc.

Minor points:

1. The authors described that TRPV2-mediated currents showed prominent outward rectification. It is described in the TRPV2 cloning paper in 1997 that TRPV2 currents show dual rectification while outward rectification is more prominent. The IV curve in Figure 1C looks dual rectification. Actually, the authors described that cells expressing mouse TRPV2 produced prominent inward as well as outward currents in page 7.

2. Page 7. The authors described that a HOAc-induced potentiation was only observed following washout of HOAc in the case of human TRPV2. The authors should discuss the mechanism for potentiation upon HOAc washout.

3. When comparing Figure 1H, Figure 3G and Figure 3H although in the different scales, this reviewer cannot recognize the big difference as shown in Figure 3I. More appropriate representative traces should be presented.

Referee #2:

The manuscript Haug et al. "Functional and structural insights into activation of TRPV2 by weak acids" provides a comprehensive analysis of acetic acid as a low affinity activator and/or positive modulator of heterologously overexpressed TRPV2 channels.

The experimental data appear sound and are described in sufficient detail. A specific strength of the paper is the cryo-EM analysis of the rTRPV2 structure and its changes upon exposure to 30 mM sodium acetate (NaAc) at pH 5.0. Based on this analysis, an alternative to NaAc binding to protonable amino acids in the outer pore region was formulated and experimentally supported by mutagenesis data. The findings are mechanistically and structurally interesting in the field of TRP channel regulation.

In previous studies, effects of the carbonic acids might have been overlooked because of the high concentrations (10-100 mM) and the highly acidic pH 5.0 that are needed to effectively modulate the channel. In the presence of extracellular Calcium ions, the effects of NaAc are partially inhibited, and stimulation with NaAc hardly induces any Ca²⁺ signal despite massive overexpression of the channel. Also, human TRPV2 appears less susceptible to this mode of activation, raising concerns about the physiological or pathophysiological relevance of the observed effects. For the broad readership of The EMBO Journal, such relevance would presumably be of utmost importance.

Major points:

1. Are there any organs, tissues or (patho)physiological conditions where such NaAc concentrations and pH values actually exist in humans or rodents? In inflamed tissues, the extracellular pH may drop to pH 5.5, but are 10-30 mM concentrations of NaAc present at the same time?

2. From Fig EV2, I understand that NaAc is only the most effective and potent modulator among the four investigated candidates. Since other organic acids (fatty acids, lactic acid, uric acid, acetoacetic acid, hydroxybutyric acid, succinic acid, fumaric acid, ...) exist as physiologically or pathophysiological relevant metabolites, NaAc may not be the most relevant member in terms of physiological or pathophysiological TRPV2 modulation.

3. Since the neutral rather than the anionic state appears to be responsible for TRPV2 modulation, the question arises whether non-deprotonatable compounds of comparable size, shape and polarity (e.g. acetaldehyde, ketone bodies) may elicit similar effects at concentrations that are reached at least in pathophysiological states.

4. The long-known TRPV2 modulator probenecid is also a carboxylic acid that strongly potentiates the effects of 2-APB. Would benzoic acid act like NaAc? If yes, do probenecid and NaAc bind to an identical/overlapping binding site?

Minor points:

From the existing on-cell and inside-out recordings, single channel properties (unitary conductance, mean open times) should be analysed, reported and discussed. Although the most likely mode of channel modulation by NaAc is given by a higher open probability, other possibilities should be considered.

Referee #3:

The authors provide insight in the weak acid dependent activation mechanisms of TRPV2. For that they provide a combined approach of mainly electrophysiology and MD simulations. Moreover, they investigate the effect of weak acid on cell cytotoxicity. The study is well performed and for sure of interest. However, before publication the authors should address the following comments:

Major comments:

The authors excluded that endogenous PAC currents are responsible for currents observed via HOA treatment. HEK293 cells express a set of other channel types, such as TRPM7, Can the authors also exclude these channels?

Inside-out currents show strong noise. What is the reason for this? Moreover, can you show single channel openings? What would happen if you whole cell you would add HOAc in the pipette? What happens if in whole cell you apply either pH5 outside or inside or at both sides?

For heat evoked currents the authors do not compare with 2-APB induced currents. In contrast, for HOAc evoked currents they compare with the effect of 2-APB. Why? Do heat-evoked currents not respond to 2-APB?

The authors report an enhancement in Glu495-Glu561 contacts compared to the simulations without protonation. But what stabilizes this Glu-Glu interplay. Shouldn't they repulse each other? The illustration of the percentage of contacts in Fig. 3J is difficult to understand. Could you please provide information on both pairs and their percentage of contacts at pH 7 and pH 5?

The authors write that "TRPV2-H521A also completely failed to generate membrane currents when exposed to 30 mM or even 100 mM HOAc at pH 5.0 (Fig. 4K-L, n=...)". However, Fig 4 K shows clear currents in the I/V relationship?

The authors show Ca²⁺ imaging experiments of TRPV2. TRPV2 is not highly Ca²⁺ selective. What about other ions, which permeate across TRPV2. How are they affected by pH and HOA?

Concerning cell death: only 30mM HOAc induced cell death. What concentration would be physiological. Could lower concentrations reduce cell growth or affect other cell pathway (e.g. NFAT, ...)?

2-APB seems to act on the same site like HOAs. In principle 2-APB should act rather only in the presence of HOA. Why is it then acting also in the absence of HOA on TRPV2 to activate it?

The manuscript would benefit from an explanatory scheme demonstrating the activation mechanism by weak acids (probably also in comparison to other activation mechanisms).

Referee #1:

Haug et al. clearly showed that weak acids such as acetic acid and lactic acid activate and sensitize TRPV2 through a mechanism requiring permeation of weak acids through the cell membrane. And the authors clarified the structural basis of the weak acids-induced TRPV2 activation with a cryo-EM and MD simulation. The presented results look very solid and support the conclusion.

This reviewer wants to ask the authors to do some experiments to strengthen their logic.

1. The authors described a mechanism requiring permeation of weak acids through the cell membrane. It would be possible to estimate the cytosolic pH changes beneath the plasma membrane upon application of weak acids to the extracellular solution. The data should be compared with the proton (inside the cell) dose-dependency for the activation of TRPV2.

Reply: We are not completely sure what the referee is asking for at this point, but we agree that it would make sense to demonstrate that the degree of HOAc-induced intracellular acidification correlates with activation of TRPV2. We already demonstrated that activation of TRPV2 by HOAc depends on concentration (10, 20 and 30 mM) as well as on pH-value (pH 7.4-6.0-5.0). We have now performed additional BCECF-based imaging recordings monitoring changes in intracellular pH induced by 10, 20 and 30 mM HoAc at pH 7.4, pH 6.0 or pH 5.0, e.g. the same conditions that were used for patch clamp experiments. As is now demonstrated in figure 2C these data perfectly correlate with the patch clamp data, e.g. the degree of intracellular acidosis induced by HOAc depends on both concentration and the pH-value. HOAc at pH 7.4 does not induce a robust intracellular acidification, and it does not gate TRPV2.

We hope that these data adequately address the point given by referee 1.

2. The authors showed the weak acid insensitivity of human TRPV2. Comparison of the TRPV2 structure with acetic acids by Cryo-EM between rat TRPV2 and human TRPV2 would support the structural basis of the weak acid-induced activation of TRPV2.

Reply: We agree with the reviewer that the structure of human TRPV2 would be very useful to have. However, from our own – not yet successful - efforts to get the structure of hTRPV2 we have learned that it is not at all a trivial task. It is not clear what makes hTRPV2 so different from other orthologues, but that is something we need to address in future studies.

3. The reviewer wants to know the changes in temperature thresholds for heat-evoked activation of rat TRPV2 in the presence of HOAc.

Reply: In the experiments exploring the effect of HOAc on heat-sensitivity of TRPV2, the applied heat did not exceed > 50°C and never induced activation of rTRPV2 in control solution. Therefore, we cannot determine the threshold for heat in untreated cells. The main reason for this approach is the pronounced “auto-sensitization” of the heat-sensitivity of rTRPV2 (DOI: 10.1016/j.bpj.2016.03.005). When activated once with a heat stimulus reaching the threshold >50°C, the channel exhibits a lower heat-threshold upon consecutive stimuli. We therefore chose to apply subthreshold heat-stimuli that do not activate TRPV2 in control solution, and then to examine the effects of these heat-stimuli on

currents induced by 10 mM HOAc at pH 5.0. Consequently, there is no clear threshold for the current potentiation that was induced by heat. We hope that our brief explanation of our approach is clear, and we have reworded the passages in the manuscript that describe these experiments.

Minor points:

1. The authors described that TRPV2-mediated currents showed prominent outward rectification. It is described in the TRPV2 cloning paper in 1997 that TRPV2 currents show dual rectification while outward rectification is more prominent. The IV curve in Figure 1C looks dual rectification. Actually, the authors described that cells expressing mouse TRPV2 produced prominent inward as well as outward currents in page 7.

Reply: We thank the reviewer for this interesting remark, we were not aware of this property of TRPV2. The reviewer is referring to one uncommented figure (1e) in the original paper from Caterina et al 1999. The current trace demonstrated in the paper stems from a heat-induced activation of rTRPV2 expressed in oocytes. We do not know how to interpret this, but we now describe the HOAc-induced membrane current in figure 1C to exhibit inward and outward rectification.

2. Page 7. The authors described that a HOAc-induced potentiation was only observed following washout of HOAc in the case of human TRPV2. The authors should discuss the mechanism for potentiation upon HOAc washout.

Reply: We have now included a brief comment on this effect in the results as well as in the discussion (page 16).

3. When comparing Figure 1H, Figure 3G and Figure 3H although in the different scales, this reviewer cannot recognize the big difference as shown in Figure 3I. More appropriate representative traces should be presented.

Reply: At this point we thank the reviewer for having identified an important mistake in figure 1! The correct label for the heat-evoked current of rTRPV2-WT in figure 1H is 2 nA, not 0.5 nA that correctly applies for figure 1I. We apologize for this mistake!

The traces in figure 3 now appear representative. For more clarity however, we have now set the label for the amplitudes to 0.4 nA for both G and H.

Referee #2:

The manuscript Haug et al. "Functional and structural insights into activation of TRPV2 by weak acids" provides a comprehensive analysis of acetic acid as a low affinity activator and/or positive modulator of heterologously overexpressed TRPV2 channels.

The experimental data appear sound and are described in sufficient detail. A specific strength of the paper is the cryo-EM analysis of the rTRPV2 structure and its changes upon exposure to 30 mM sodium acetate (NaAc) at pH 5.0. Based on this analysis, an alternative to NaAc binding to protonable amino acids in the outer pore region was formulated and experimentally supported by mutagenesis data. The findings are mechanistically and structurally interesting in the field of TRP channel regulation.

In previous studies, effects of the carbonic acids might have been overlooked because of the high concentrations (10-100 mM) and the highly acidic pH 5.0 that are needed to effectively modulate the channel. In the presence of extracellular Calcium ions, the effects of NaAc are partially inhibited, and stimulation with NaAc hardly induces any Ca²⁺ signal despite massive overexpression of the channel. Also, human TRPV2 appears less susceptible to this mode of activation, raising concerns about the physiological or pathophysiological relevance of the observed effects. For the broad readership of The EMBO Journal, such relevance would presumably be of utmost importance.

Major points:

1. Are there any organs, tissues or (patho)physiological conditions where such NaAc concentrations and pH values actually exist in humans or rodents? In inflamed tissues, the extracellular pH may drop to pH 5.5, but are 10-30 mM concentrations of NaAc present at the same time?

Reply: We have been thinking and discussing at lot along similar lines, e.g. which may be the physiological conditions at which high concentrations of weak acids with strong acidosis occur? While there are several possible scenarios, we only have limited space to speculate about this in the discussion. We have now extended the discussion on this point (page 17), but in the end this is a mechanistic study that we should not overuse for speculation.

2. From Fig EV2, I understand that NaAc is only the most effective and potent modulator among the four investigated candidates. Since other organic acids (fatty acids, lactic acid, uric acid, acetoacetic acid, hydroxybutyric acid, succinic acid, fumaric acid, ...) exist as physiologically or pathophysiologicaly relevant metabolites, NaAc may not be the most relevant member in terms of physiological or pathophysiological TRPV2 modulation.

Reply: We fully agree that HoAc may not be the most relevant weak acid in terms in physiological relevance, but it was the most effective and feasible acid for probing the general mechanism. Saying this, our study do contain data showing that several other weak acids activate TRPV2 as well, including lactic acid and CO₂. There may be several further relevant weak acids that we need to examine in this regard, but for now we argue that just adding several further weak acids would probably not strengthen this manuscript to a great extent.

3. Since the neutral rather than the anionic state appears to be responsible for TRPV2 modulation, the question arises whether non-deprotonatable compounds of comparable size, shape and polarity (e.g. acetaldehyde, ketone bodies) may elicit similar effects at concentrations that are reached at least in pathophysiological states.

Reply: We have now examined the effects of the ketonic body β -hydroxybutyric acid on rTRPV2, and we observed absolutely no effects, Appencic 1K-L. A previous reports observed an activation of TRPA1 by acetaldehyde, but TRPV2 was insensitive doi: 10.1111/j.1460-9568.2007.05882.x. We have confirmed this result, but as it was already published we have not included these data in the manuscript.

4. The long-known TRPV2 modulator probenecid is also a carboxylic acid that stronlgy potentiates the effects of 2-APB. Would benzoic acid act like NaAc? If yes, do probenecid and NaAc bind to an identical/overlapping binding site?

Reply: The reviewer raises an intriguing point. When it comes to benzoic acid, we do see both activation and sensitization of TRPV2. These data have now included in Fig. EV2P- S. We are currently performing a separate and quite extensive study on the effects of probenecid on TRPV2, including the role of probenecid as a weak acid. That story is well beyond the scope of the current study, so we chose not to address probenecid-induced activation of TRPV2 in this manuscript.

During the preparation of this revision a report describing a very potent activation of TRPV2 by probenecid and several derivatives of benzoic acid was published (doi: 10.1016/j.csbj.2023.12.028.). We now cite this report in the manuscript, and we have also included a brief comment on the proposed binding site on TRPV2 for probenecid in the discussion (Page 15).

Minor points:

From the existing on-cell and inside-out recordings, single channel properties (unitary conductance, mean open times) should be analysed, reported and discussed. Although the most likely mode of channel modulation by NaAc is given by a higher open probability, other possibilities should be considered.

Reply: The already included data from on-cell and inside-out experiments were not designed for a more detailed analysis of single channel properties (filtering etc). We have now performed additional inside-out experiments on rTRPV2, displaying single channel openings induced by HOAc, and in the same patches also by 2-APB. The data clearly indicate that weak acids increase open probability of TRPV2. Please see figure 2H- K as well as Appendix Figure 2.

We looked into the literature to review what is generally known about single channel properties of TRPV2. We were surprised to note that very little has been done, thus only few papers present very limited data on single channel properties of TRPV2. Thus, a more detailed analysis of single channel properties with weak acids and other agonists is warranted, but it would go beyond the scope of this study.

Referee #3:

The authors provide insight in the weak acid dependent activation mechanisms of TRPV2. For that they provide a combined approach of mainly electrophysiology and MD simulations. Moreover, they investigate the effect of weak acid on cell cytotoxicity. The study is well performed and for sure of interest. However, before publication the authors should address the following comments:

Major comments:

The authors excluded that endogenous PAC currents are responsible for currents observed via HOA treatment. HEK293 cells express a set of other channel types, such as TRPM7, Can the authors also exclude these channels?

Reply: We were not aware of an endogenous expression of TRPM7 in HEK 293 cells. As we do not see any effects of weak acids on non-transfected cells however, we can almost rule out that TRPM7 may contribute to membrane currents observed in this study. Several reports on TRPM7 convincingly show that the channel is inhibited by intracellular acidosis and thus also by weak acids(doi:

10.1074/jbc.RA118.004066.; doi: 10.1085/jgp.200509324). So if anything, endogenous TRPM7 channels may be inhibited in our experiments.

Inside-out currents show strong noise. What is the reason for this? Moreover, can you show single channel openings? What would happen if you whole cell you would add HOAc in the pipette? What happens if in whole cell you apply either pH5 outside or inside or at both sides?

Reply: Please see our reply to reviewer 2 above, we have now included data on single channel recordings.

We have tried many different approaches to gain further insights into the mechanism of TRPV2-activation by weak acids, including experiments with HOAc and pH 5 in the pipette. However, the experiments gave inconsistent data, and in case of "pH 5 inside and outside" we sometimes observe activation of PAC-like currents. As these data would not add anything of value, we did not further pursue this approach.

For heat evoked currents the authors do not compare with 2-APB induced currents. In contrast, for HOAc evoked currents they compare with the effect of 2-APB. Why? Do heat-evoked currents not respond to 2-APB?

Reply: For all mutants, we verified their functionality by applying 2-APB at the high concentration of 1 mM. This was done in the experiments examining direct activation induced by 30 HOAc at pH 5.0. This approach also allowed us to normalize sensitivity to HOAc against 2-APB, approach enabling us to separate non-functional mutants against mutants with a more specific loss of weak acid sensitivity.

When it comes to the experiments examining heat-evoked currents, an additional application of 2-APB would not enable us to draw more definite conclusions. For sure, application of 2-APB would cause a very strong potentiation of heat-evoked currents as well, but we did not perform such experiments.

The authors report an enhancement in Glu495-Glu561 contacts compared to the simulations without protonation. But what stabilizes this Glu-Glu interplay. Shouldn't they repulse each other? The illustration of the percentage of contacts in Fig. 3J is difficult to understand. Could you please provide information on both pairs and their percentage of contacts at pH 7 and pH 5?

Reply: Based on the PROPKA predicted pKas for these residues in the two different conformations, these residues have a moderate to high likelihood of being protonated at pH 5. This could allow for the formation of carboxyl-carboxylate interactions between the two glutamic acid residues, which is consistent with the close proximity for Glu495 and Glu561 we observe in the TRPV2_{HOAc} structure and in the simulation where these residues are protonated (simulating pH 5).

We have now changed Fig. 3J to separate the bars for each contact, which we hope clarifies the figure.

We have added a line in the results section to discuss the carboxyl-carboxylate interaction and have also added the percentage of contacts to the text in the results section.

The authors write that "TRPV2-H521A also completely failed to generate membrane currents when

exposed to 30 mM or even 100 mM HOAc at pH 5.0 (Fig. 4K-L, n=...)". However, Fig 4 K shows clear currents in the I/V relationship?

Reply: We agree that the current trace demonstrated in figure 4K is somewhat confusing due to a leak current. We have inserted a more representative current trace with a smaller leak current, and from that current it becomes clear that the TRPV2-H521A mutant is not activated by HOAc.

The authors show Ca²⁺ imaging experiments of TRPV2. TRPV2 is not highly Ca²⁺ selective. What about other ions, which permeate across TRPV2. How are they affected by pH and HOA?

Reply: This question can give rise to several experiments. In initial experiments we indeed tried to perform Na⁺ imaging with weak acids on TRPV2. However, the signal of the fluorescent dye was strongly quenched by weak acids, thus we could not pursue that approach. We did not specifically look at other ions.

TRPV2 is commonly referred to as a channel with a high calcium permeability, the only citation that can be used for this note is the original paper from Caterina et al., 1999. However, the ion permeability of TRPV2 was only determined for activation by high heat (> 52°C) in that study. To our surprise, we cannot find any further studies showing a high calcium permeability for TRPV2 when activated by other agonists. Therefore, we have now performed experiments to determine the permeability to calcium and magnesium upon activation by 2-APB, CBD and HOAc. As is now demonstrated in figure 5L, we found that rTRPV2 displays a high permeability for both calcium and magnesium when activated by 2-APB and CBD. The relative permeabilities were indeed similar to those reported by Caterina et al., 1999. When activated by HOAc however, the permeabilities to both calcium and magnesium were considerably lower.

Concerning cell death: only 30mM HOAc induced cell death. What concentration would be physiological. Could lower concentrations reduce cell growth or affect other cell pathway (e.g. NFAT, ...)?

Reply: This is an important question and we are aware that our experiments on HOAc-induced cell death fall short in several details, including concentration-dependency and involved intracellular signaling pathways. However, we feel that a more detailed elaboration on this property should be performed in a separate report allowing enough room to demonstrate conclusive data on a complex property. Nevertheless, our data indicate that TRPV2 can mediate cell injury when challenged with weak acids.

2-APB seems to act on the same site like HOAs. In principle 2-APB should act rather only in the presence of HOA. Why is it then acting also in the absence of HOA on TRPV2 to activate it?

Reply: 2-APB has been a well-established activator of TRPV2 for several years now and while acid may enhance the effect of 2-APB, either through potentially altering the structure of 2-APB (doi: [10.1038/srep20791](https://doi.org/10.1038/srep20791)) or sensitizing the channel as we show here, it is not necessary for channel activation.

We previously demonstrated that 2-APB-sensitivity of rTRPV2 involves H521, R535 and R539, e.g. the same residues that also dictate sensitivity to weak acids. However, replacing these residues does not abolish 2-APB-sensitivity, but it results in a strongly reduced sensitivity (doi: [10.1038/s41467-022-](https://doi.org/10.1038/s41467-022-)

30083-3). Thus the mechanism for activation by 2-APB seems to be more complex, and it might involve further binding sites as was recently suggested (doi: 10.1038/s41589-022-01139-8). 2-APB also activates TRPV1, TRPV3 and TRPA1, and the binding sites are not conserved among these channels. TRPV1 is highly 2-APB-sensitive, but it even gets inhibited by weak acids. Therefore, 2-APB-sensitivity does not have to go along with weak acid-sensitivity.

The manuscript would benefit from an explanatory scheme demonstrating the activation mechanism by weak acids (probably also in comparison to other activation mechanisms).

Reply: We have included a simplistic model in figure 2.

Dear Dr Moiseenkova-Bell, dear Dr Leffler,

Thank you for submitting your revised manuscript (EMBOJ-2023-115715R) to The EMBO Journal. Your amended study was sent back to the three referees for their scientific re-evaluation, and we have received detailed comments from two of them, which I enclose below.

Please note that while reviewer #1 was at this time not able to reassess your revised manuscript we have carefully assessed your response to the critique raised by this expert editorially and found the issues to be addressed satisfactorily. Specifically, we concluded that the request for integration of the current results with additional human structural data (ref#1, pt.2) is well taken as such but goes beyond the scope of the current work in our view.

As you will see, the other two experts state that the work has been substantially improved by the revisions and they are now broadly in favour of publication.

Thus, we are pleased to inform you that your manuscript has been accepted in principle for publication in The EMBO Journal.

We now need you to take care of a number of issues related to formatting and data presentation as detailed below, which should be addressed at re-submission.

Please contact me at any time if you have additional questions related to below points.

As you might have seen on our web page, every paper at the EMBO Journal now includes a 'Synopsis', displayed on the html and freely accessible to all readers. The synopsis includes a 'model' figure as well as 2-5 one-short-sentence bullet points that summarize the article. I would appreciate if you could provide this figure and the bullet points.

Thank you for giving us the chance to consider your manuscript for The EMBO Journal. I look forward to your final revision.

Again, please contact me at any time if you need any help or have further questions.

Best regards,

Daniel Klimmeck

>> Author Contributions: Please remove the author contributions information from the manuscript text. Note that CRediT has replaced the traditional author contributions section as of now because it offers a systematic machine-readable author contributions format that allows for more effective research assessment. and use the free text boxes beneath each contributing author's name to add specific details on the author's contribution.

More information is available in our guide to authors.
<https://www.embopress.org/page/journal/14602075/authorguide>

>> Adjust the title of the 'Competing Interest Statement' section to 'Disclosure and Competing Interests Statement' and move after Acknowledgements.

>> Funding: please enter 'the Swedish e-Science Research Centre and the Friedrich und Alida Gehrke-Stiftung' into our online manuscript system.

>> References: adjust reference format to 10 authors et al, and place References after the Discussion, before figure legends.

>> Appendix: the appendix requires a ToC on its first page. Nomenclature of figures and tables should be adjusted to "Appendix Figure S1, S2..." "Appendix Table S1" etc. Please correct textual callouts accordingly.

>>Appendix Figures 4 and 6 appear to be identical. please check and substitute. Change figure orientation to portrait.

>>Main figure dimensions are not standard (A4). Please ensure the figures adhere to the journal guidelines.

https://www.embopress.org/pb-assets/embo-site/EMBOPress_Figure_Guidelines_061115-1561436025777.pdf

>> Source Data: needs to be provided for Fig 1H, Fig 4E, Fig 6A-D.

>>Author Checklist: please correct the manuscript number in the top header.

>> Add the Reagents and tools table to the manuscript text, at the beginning of Materials & Methods.

>> Consider additional changes and comments from our production team as indicated below:

- Figure legends:

1. Please note that a separate 'Data Information' section is required in the legends of figures 4d, g.
2. Please note that the legend for figure 5j is incorrectly labelled as 5l in the manuscript. This needs to be rectified.
3. Please note that the figure title is missing in the legends for figures EV1-5.
4. Please define the annotated p values ***/**/* in the legends of figures 3f; 4j; as appropriate.
5. Please indicate the statistical test used for data analysis in the legends of figures 1l; 3f, i; 4j, n; 5d, g; 6e.
6. Please note that in figure 3i there is a mismatch between the annotated p values in the figure legend and the annotated p values in the figure file that should be corrected.
7. Please note that the box plots need to be defined in terms of minima, maxima, centre, bounds of box and whiskers, and percentile in the legends of figures 1l; 2b, f, j-k; 3f, i; 4j, n; 5d, g; 6e; EV 1d, f, k; EV 2b, d, f, h, j, l, n, q, s; EV 3a-b, f, i.
8. Please note that the box plot needs to be defined in terms of centre, bounds of box and percentile in the legend of figure 5j.
9. Please note that information related to n is missing in the legends of figures 1e, l; 2b, f, j-k; 3f, i; 4j, n; 5d, g, j; 6e; EV 1d, f, h, k; EV 2b, d, f, h, j, l, n, q, s; EV 3a-c, f, i-j.
10. Please note that the error bars are not defined in the legend of figure 1e.
11. Please note that scale bar and its definition are missing for figures 6a-d.

We realize that it is difficult to revise to a specific deadline. In the interest of protecting the conceptual advance provided by the work, we recommend a revision within 3 months (6th Jun 2024). Please discuss the revision progress ahead of this time with the editor if you require more time to complete the revisions.

Referee #2:

The revised version of the manuscript has been considerably improved by adding new data and discussing their implications.

I specifically appreciate the single channel data, which is both interesting and intriguing since single channel properties seem not to be identical for different activators and settings. This is reminiscent of data obtained for TRPV3, which shows variable single channel conductances depending on the agonist as well.

With no further critique, I judge the manuscript suitable for publication in the EMBO Journal.

Referee #3:

The authors have addressed all reviewers comments satisfactory.

The authors addressed the minor editorial issues.

Dear Dr Leffler,

Thank you for submitting the revised version of your manuscript. I have now evaluated your amended manuscript and concluded that the remaining minor concerns have been sufficiently addressed.

I am pleased to inform you that your manuscript has been accepted for publication in the EMBO Journal.

On a different note, I would like to alert you that EMBO Press offers a format for a video-synopsis of work published with us, which essentially is a short, author-generated film explaining the core findings in hand drawings, and, as we believe, can be very useful to increase visibility of the work. Please see the following link for representative examples and their integration into the article web page:

<https://www.embopress.org/doi/full/10.15252/emboj.2019103932>

Best regards,

Daniel Klimmeck

Daniel Klimmeck, PhD
Senior Editor
The EMBO Journal
EMBO
Postfach 1022-40
Meyerhofstrasse 1
D-69117 Heidelberg
contact@embojournal.org
Submit at: <http://emboj.msubmit.net>
